

# The response of supraglacial debris to elevated, high frequency GPR: Volumetric scatter and interfacial dielectric contrasts interpreted from field and experimental studies

Alexandra Giese[1], Steven Arcone[2], Robert Hawley[1], Gabriel Lewis[1], and Patrick Wagnon[3]

[1]Department of Earth Sciences, Dartmouth College, Hanover, NH USA
[2]Thayer School of Engineering, Dartmouth College, Hanover, NH USA
[3]Univ. Grenoble Alpes, CNRS, IRD, Grenoble INP, IGE, F-38000 Grenoble, France

**Correspondence:** Robert Hawley (robert.l.hawley@dartmouth.edu)

**Abstract.** The thickness of supraglacial debris affects the surface energy balance and retreat patterns of mountain glaciers. Therefore, knowing a debris layer's thickness is crucial for understanding the magnitude and timeframe of glacier melt. Field-based ground-penetrating radar (GPR) has recently gained attention as a possible method for measuring debris thickness. Airborne assessments achieve extensive coverage, but the use of GPR for such platforms remains relatively unexplored. We

5  investigated the performance of 960 MHz and 2.6 GHz GPR signals through dry laboratory rock debris, and of 960 MHz over $\sim 2$ km of transects on the debris cover of Changri Nup Glacier, Nepal Himalaya. On the glacier, 960 MHz profiles were characterized by volumetric backscatter from within $\sim 10 - 40$ cm, a depth that corresponds to approximate ground-truth debris thicknesses on all transects, with no clear reflection from the ice interface. The laboratory results show that the lack of an ice-debris interface return in field data was likely caused by a weak dielectric contrast between solid ice and porous dry debris and that surface scatter is coherent but weak. This suggests that the debris-ice interface reflection was also likely

10  coherent, supporting our conclusion of a weak dielectric contrast. The laboratory 2.6 GHz results show significant penetration for only smaller clast sizes up to 4 cm. We used a statistical approach to estimate ice depths from volumetric scatter, which gave reasonable agreement with ground-truth depth measurements. We conclude that a remote system operating near 1 GHz could successfully estimate dry debris cover thicknesses based on depth of volumetric backscatter.

## 1 Introduction

In High Mountain Asia (HMA), glacier melt contributes to rivers that supply water to a significant portion of the global population (Kraaijenbrink et al., 2017; Rowan et al., 2017; Brun et al., 2017). Debris covers 14-18% of Himalayan glacier area (Kääb et al., 2012), compared to a worldwide figure of only 4.4% (Scherler et al., 2018). The share of glacier area covered in

debris is even greater eastward, at 25% in East Nepal (Kraaijenbrink et al., 2017) and 36% in the Everest region (Thakuri et al., 2014). Debris layers form in a glacier's ablation zone and are, consequently, highly relevant to glacier melt. Understanding the impact of climate change on Himalayan glaciers is critical to predicting the future water supply of a highly populated region (Lutz et al., 2014; Rowan et al., 2017). The response of debris covered glaciers to climate change is critically important, because the influence of debris is projected to increase in a warming climate (Scherler et al., 2018; Kraaijenbrink et al., 2017).

Thickness of debris on a glacier is of paramount importance because it affects ablation, and thus measurements of debris thinckness are needed. Here we investigate how high-frequency ground-penetrating radar (GPR) signals from backscatter may indicate debris thickness.

Debris is rockfall from valley walls (Menzies and van der Meer, 2017). After englacial transport, material deposited in the accumulation zone resurfaces in the ablation area (Rowan et al., 2015); clast shapes and degrees of angularity vary widely, and

abrupt mineralogy transitions may occur given a range of rock sources. Exposed glacial debris is loosely packed, highly porous, erratically bedded, and variable in thickness over meter to decimeter spatial scales. At any location there is a random distribution of sizes and a lack of sorting beyond that which occurs during supraglacial resedimentation (Lawson, 1979; Menzies and van der Meer, 2017). On the glacier scale, debris generally increases in thickness towards the terminus, as more and more debris accumulates and resurfaces. High Mountain Asia debris layer thickness ranges from a few mm at the upglacier start of

the debris layer to several meters (Ragettli et al., 2015; McCarthy et al., 2017; Rowan et al., 2017) at the glacier terminus, and individual debris clasts span fine sand to boulders exceeding 10 m.

As shown through experiments (Östrem, 1959; Reznichenko et al., 2010) and *in situ* field measurements (Mattson et al., 1993; Nicholson and Benn, 2006), a debris-covered glacier's response to climate depends on debris thickness. Where less than a few cm, debris enhances underlying glacier melt because it has a lower albedo relative to ice and snow. Over a few

cm thick, debris insulates. The thickness dependence of melt has informed development of many models (e.g. Nicholson and Benn, 2006; Reid and Brock, 2010), and Evatt et al. (2015) simulated the peak melt at a few cm. Because debris thickness is a primary control on sub-debris ablation, accurate measurement of its thickness is paramount for projections of glacial melt and the associated timing and quantity of fresh water. Further, the spatial variability of debris is known to be large (e.g. Reid et al., 2012; Nicholson and Mertes, 2017) but is, nevertheless, essential for assessing glacier-scale melt trends.

Historically, debris thickness has been determined by manual excavations (e.g. Zhang et al., 2011). Other approaches include geometrical scaling estimations of exposed debris (Nicholson and Benn, 2012; Nicholson and Mertes, 2017) and calculations from energy balance models in concert with remote thermal imagery (e.g. Foster et al., 2012; Rounce and McKinney, 2014; Schauwecker et al., 2015) or surface height changes (Ragettli et al., 2015). Empirical relationships between thickness and



remotely detected surface temperature have been derived (Juen et al., 2014; Mihalcea et al., 2006, 2008), and debris thickness has also been computed from elevation change and flux divergence by inverting a melt model (Rounce et al., 2018).

Recent studies have employed ground-based GPR on supraglacial debris to determine thickness (e.g. Mackay et al. (2014) in Antarctica; Wu and Liu (2012) in China's Tien Shan). McCarthy et al. (2017) used 225, 450, 900, and 1200 MHz GPR to
observe continuous horizons from debris-ice interfaces in 16 of 29 profiles covering $\sim 600$ m on Lirung Glacier, Nepal. They characterized the electromagnetic (EM) returns from the debris as having high scatter relative to the ice reflections. Nicholson and Mertes (2017) used 200/600 MHz dual frequency GPR to validate thickness calculations on Ngozumpa glacier (Nepal), and Nicholson et al. (2018) used the same dual frequency GPR on the same glacier to collect $\sim 3.5$ km of radar profiles. The ice surface was distinct enough in the majority of the profiles to be picked manually, but a distinct debris-ice interface is not
always present. Wu and Liu (2012) and Nicholson et al. (2018) used low frequencies, irrelevant for efficient areal coverage, and collected continuous data with dragged antennas, an approach that was impossible at our field site. Wu and Liu (2012) report an $\epsilon_{debris} = 12 - 30$, implying a wet layer. McCarthy et al. (2017)'s velocity of $0.118$ m/ns gives an $\epsilon_{debris} = 6.46$, implying a layer of solid granite or very dense debris.

On a wider spatial scale, Huang et al. (2017) used 1.27 GHz satellite Synthetic Aperture Radar (SAR) to isolate volume
scattering power caused by debris cover on Koxcar Glacier, Tienshan, China, which they inverted for debris thickness by decomposing signals into various glacier targets. Huang et al. (2017)'s results agreed well with published thicknesses near the terminus, with the debris layer primarily composed of gravels and coarse sand with 12% porosity.

Given a debris layer's heterogeneity, however, the surface and volume of debris may significantly scatter radar waves when wavelengths are comparable to or smaller than average clast size. Additionally, variable unfrozen water content could greatly
affect local relative dielectric permittivity, making depth interpretation difficult. Although we did not observe a clear reflection from the debris-ice interface, yet we note that studies have succeeded in penetrating porous, rounded rocks in the laboratory (Liu et al., 2013) and embedded within glacial till (Arcone et al., 2014). Accordingly, we hypothesized that where backscatter from angular clasts is favorable, volumetric backscatter itself, significant at frequencies useful for airborne surveys, may indicate debris depth. No previous work has directly addressed volumetric scatter of larger-clast debris like that characterizing Changri
Nup.

Our aim was to find how a frequency relevant to remote radar systems (i.e.$\sim 1$ GHz) performs in glacial debris. To this end, we compared the depth of volumetric backscatter from GPR data with ground truth measurements of debris thickness. We validate our indirect backscatter method with experimental studies.

## 2   Study Area and Methods

### 2.1   Field Site: Changri Nup Glacier

The $\sim 4$ km long Changri Nup Glacier ($27.987^o$ N, $86.785^o$ E), in the Nepal Himalaya (Figure 1), flows southeasterly from above 5700 to its terminus at 5240 m (Vincent et al., 2016). It terminates on land short of the Khumbu glacier in the Mt.





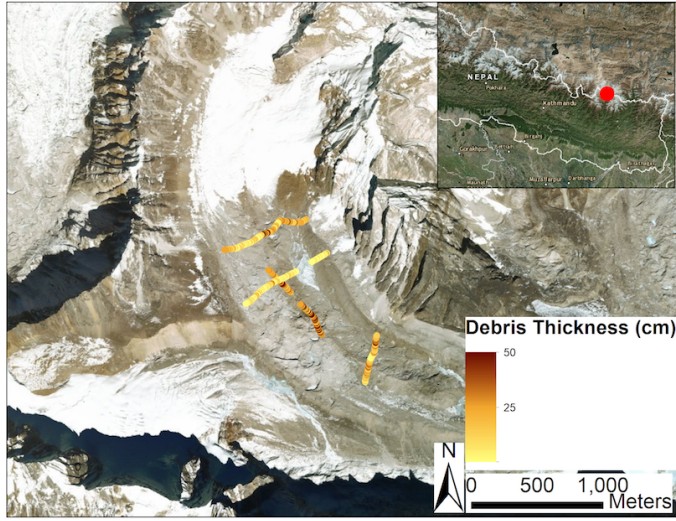

**Figure 1.** GPR profiles taken on Changri Nup Glacier with a 960 MHz antenna. Colorbar indicates debris depth determined by volumetric backscatter in GPR profiles.

Everest/Sagarmatha region (Figure 1 inset). Its lower reaches are covered by a debris layer 2.3 km long by 0.7 km wide (Vincent et al., 2016).

Debris on Changri Nup matches the surrounding bedrock in its lithology: granite, pegmatite, gneiss, pelite, calc-silicate, and amphibolite, with minerals including K-feldspar, quartz, biotite, muscovite, hornblende, plagioclase, and sillimanite (Searle et al., 2003). Changri Nup's ablation zone has a nearly continuous debris mantle, punctuated by ice cliffs, surface ponds, and a large ice sail. Example surface features are shown in Figure 2.

## 2.2 GPR and other field measurements

We used Geophysical Survey Systems, Inc.'s SIR 3000 control unit and Model 3101D "900" MHz antenna unit (pulse dominant wavelength 960 MHz). We collected five GPR profiles on three transverse (Q, S, W) and 2 longitudinal (cross-ZA, cross-PS) transects shown in Figure 1 and enumerated in Table 1 from November 14 - 27, 2015. This timing, after the monsoon and before the winter cyclones, minimized the presence of wet debris. The profiles were spatially distributed over the debris layer of Changri Nup, though longitudinal profiling was limited in favor of collecting more thickness data in the upper ablation zone. Profile names and locations follow Vincent et al. (2015) and Vincent et al. (2016). We located transect end points and recorded our paths using a handheld Garmin76 GPS, which has a manufacturer stated accuracy of $\pm 15$ m.



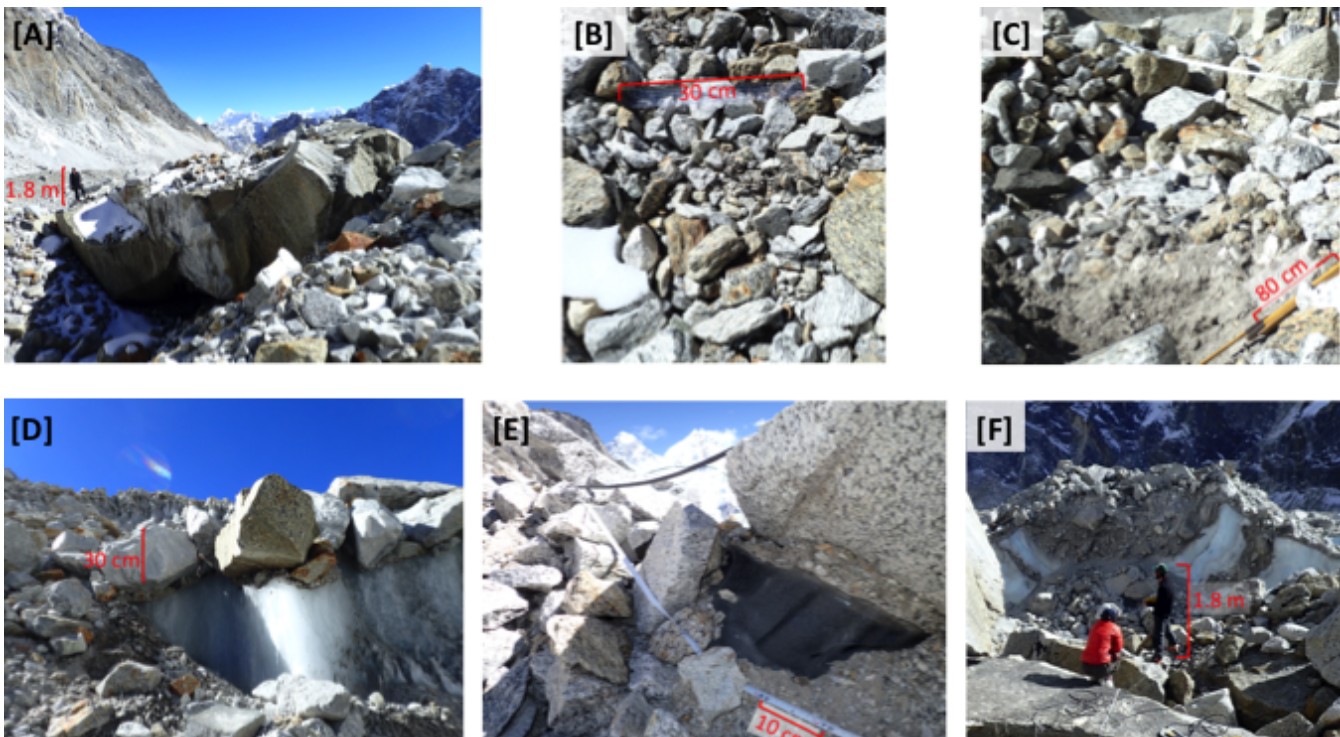

**Figure 2.** Pictures of the felsic granitic and metamorphic debris layer on Changri Nup, with scales marked. (a,b) Clasts are angular and range in size from fine sand to boulders $\sim 10$ m in diameter, and at any location there is a random distribution of sizes. (c) The debris shows inverse grading, with finer particulates buried deeper and closer to the debris-ice interface, as a result of supraglacial resedimentation (Lawson, 1979; Menzies and van der Meer, 2017). (d) Larger clasts, nested in or superimposed on the sandy boulder-gravel, are monolithic (Hambrey et al., 2008). Exposed ice reveals (e) differential ablation and (f) meltout of englacial debris.

| Transect | Length (m) | Antenna Height (cm) |
|----------|------------|---------------------|
| Q | 700 | 19 |
| S | 600 | 27 |
| W | 400 | 19 |
| Cross-ZA | 140 | 19 |
| Cross-PS | 200 | 19 |

**Table 1.** Details about the five 960 MHz GPR profiles recorded on Changri Nup Glacier in 2015. When converting time delays to target depth, we added 1 cm to antenna height to account for the 16 cm separation of the transmit and receive antennas.



A 960 MHz antenna transmits a pulse with dominant wavelengths of $\sim 31$ cm in air and $\sim 18$ cm in debris with a relative permittivity ($\epsilon$) of 3. We recorded 1024 samples at 16-bit resolution in a 50 ns time range for each trace. Logistics of profiling over a very rough surface required recording traces every 10 cm in single point mode; subsequent 10x stacking provided a 1 m horizontal resolution. The 960 MHz antenna unit was raised by 27 cm (transect S) or 19 cm (other transects) to place the

5 ground surface in the antenna's far-field. Though as shown later this elevation caused the antenna direct coupling (DC) signal to interfere with the return from the debris surface, the closer placement enhanced our chances of detecting an ice-interface signal.

Our postprocessing used RADAN 7.0 software and included resetting time zero to minimize dead time, stacking and band-pass filtering to reduce noise, horizontal filtering to remove the clutter of antenna reflections, linear range gain to compensate

10 signal loss caused by spherical beam spread, spiking deconvolution to reduce signal duration, and Hilbert magnitude transform to capture positive (in sign) energy envelopes of returns (Arcone, 1996; Yilmaz, 1987).

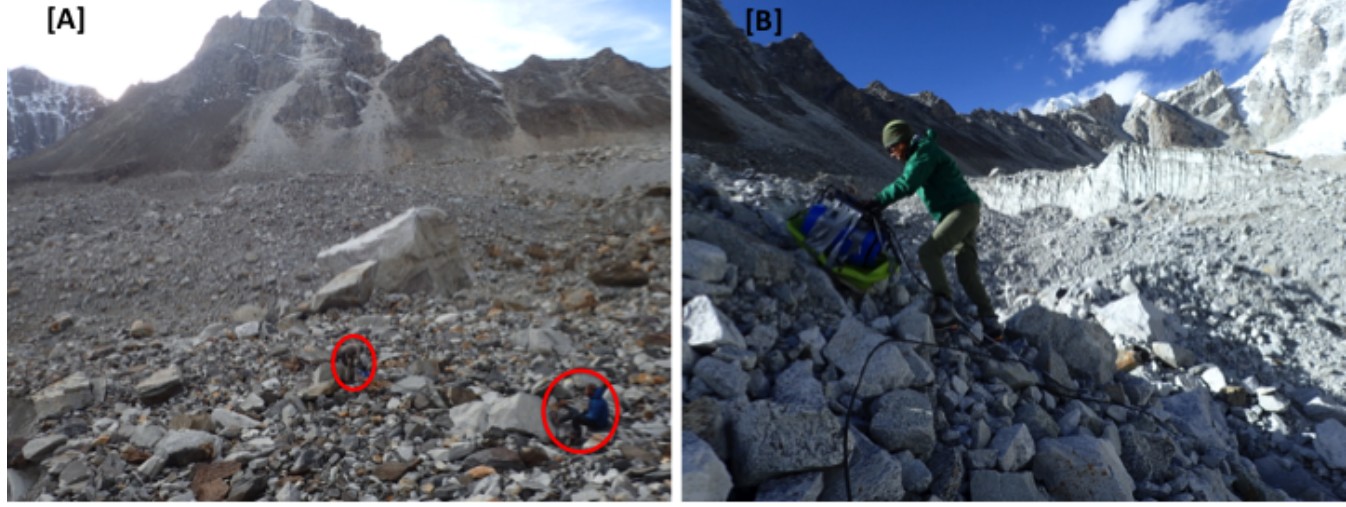

**Figure 3.** Antenna operator, shown collecting data on (a) transect Q with a 19-cm-elevated antenna and (b) transect S, which was the steepest, with a 27-cm-elevated antenna.

The antenna operator was always adjacent to the antenna (Figure 3); subsequent tests confirmed no interference from body reflections. The face of the antennas was generally kept in approximate alignment with the surface. Though the surface was not flat, most deviations were within $\pm 30^{o}$ from horizontal. These deviations and associated changes in polarization relative





to transect direction did not impact results because the antenna has a wide $70^o$ 3-dB two-way beamwidth in air regardless of polarization (Arcone et al., 1986).

We made detailed observations of ground debris, including 353 measurements of thickness made by digging to the debris-ice interface. Sizes varied widely, and debris clasts were generally very jagged and angular. The majority of clasts were less than
one 960 MHz wavelength in air (31 cm), while an estimated one-half to two-thirds of debris volume was comprised of rock fragments less than the wavelength in debris (18 cm). The bulk (94%) of ground truth measurements via excavation reached a distinct glacier surface. In the remaining 6% of measurements, the debris was frozen, and excavation was impossible beyond the recorded depth. Although the glacier surface is distinct (Figure 2d), there are clasts embedded in the ice (Figure 2e). Since much of the debris is transported englacially before melting out (Figure 2f), the ice includes embedded grains. The ice surface
where observed appeared smooth, although the sub-debris glacier surface topography is unknown (Nicholson and Mertes, 2017).

## 2.3  Rock Box Experiments

To aid analysis and interpretation of Changri Nup data, we recorded profiles with 960 MHz and 2.6 GHz antenna units over a Changri Nup Glacier analog in terms of the clast mineralogy and shape, and sub-debris interface reflectivity. Freshly broken
rocks sourced from a local quarry were angular and faceted quartzo-feldspathic gray gneiss, similar to the debris layer on Changri Nup Glacier. The rocks were placed in a trough (Figure 4) in three clast size groups, small ($1 - 4$ cm), medium ($5 - 9$ cm), and large ($10 - 20$ cm), over 4 cm thick dry pine ($\epsilon = 1.9 - 2.0$) boards above densely packed pine shavings ($\epsilon = 1.4$). We initially assumed $\epsilon_{debris} = \epsilon_{sand} = 2.6$, and the debris-pine boundary would have provided a reflectivity of approximately 0.07, similar to but slightly greater than that of similar debris above solid glacier ice (0.05); more exact measurements discussed
later provided $\epsilon$ near 3. We profiled with both antennas at $\sim 50$ cm height with and without a sheet of aluminum foil at the base of the rocks; the foil allowed straightforward $\epsilon$ calibration from the time delay through the three rock sections, while the offset allowed us to separate any surface reflection from the DC. Using velocity $\equiv c/\sqrt{\epsilon}$, where $c$ is the speed of light, we converted signal return time into depth.

## 3  Results and Analysis

### 3.1  Changri Nup Glacier

We collected each of the five profiles (Figure 1) over varied surface topographies (see Appendix A for images of the terrain along each transect). Figure 5 shows a sample 75 m section along transect S, before processing. Figure 6 shows a sample trace with various events labeled, and its representation after Hilbert magnitude transformation. Of particular importance is the overlap of the DC between antannas and the surface reflection. Typical returned pulse spectra centered between $900 - 1000$
MHz and is shown in Figure 7, in which it is evident that the subsurface volume scatter (vs) is centered near the transmitted dominant frequency of 960 MHz.




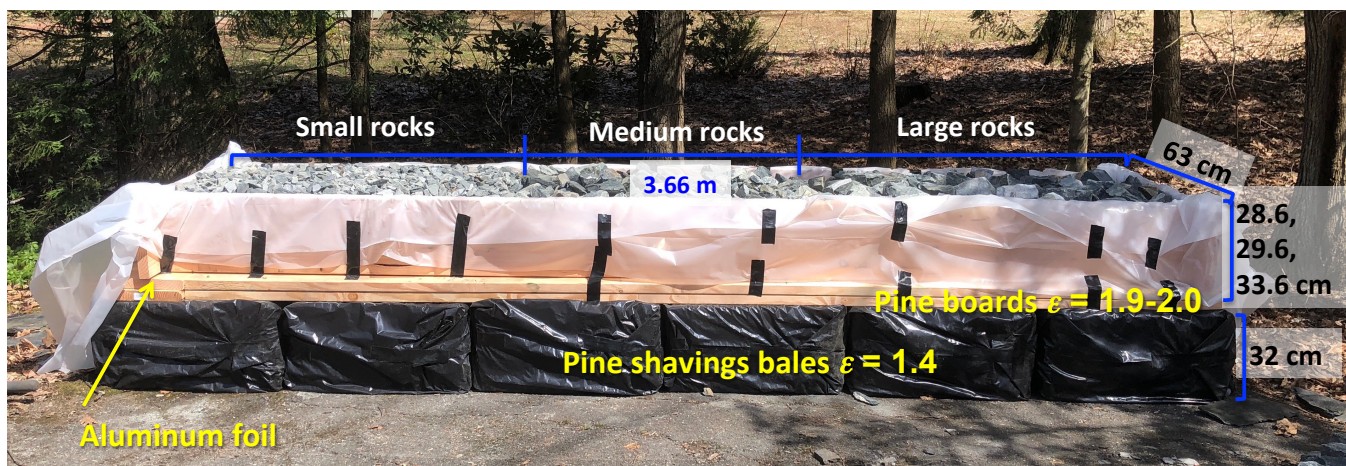

(a)

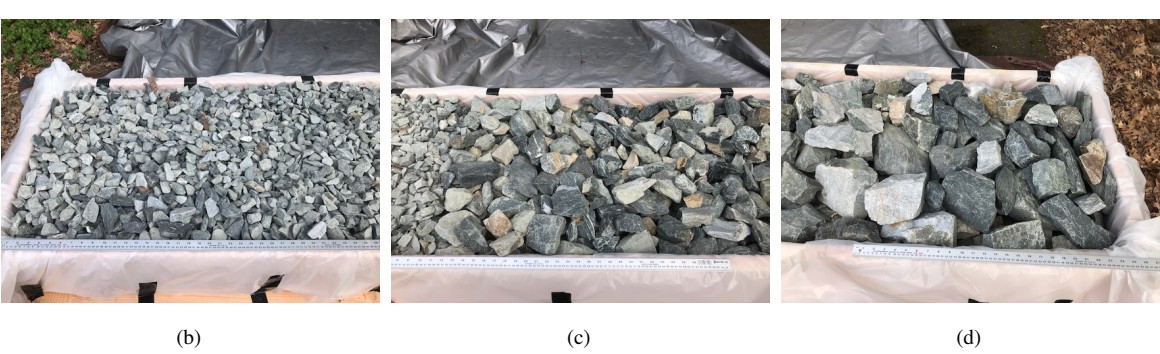

(b)                          (c)                          (d)

**Figure 4.** (a) Schematic of trough with rocks of three clast sizes resting on 4 cm thick dry pine boards over bales of pine shavings. The experimental materials were selected to simulate the reflection magnitude of ice underlying dry gravel debris. The board - bale interface reflectivity is insignificant. Aluminum foil placed beneath the rocks confirmed profile placement of the debris bottom and, thus, aided determination of wave velocity. The rocks protruded above the 28.6 cm high board edge by 0 cm (small, (b)), 1 cm (medium, (c)), 5 cm (large, (d)).



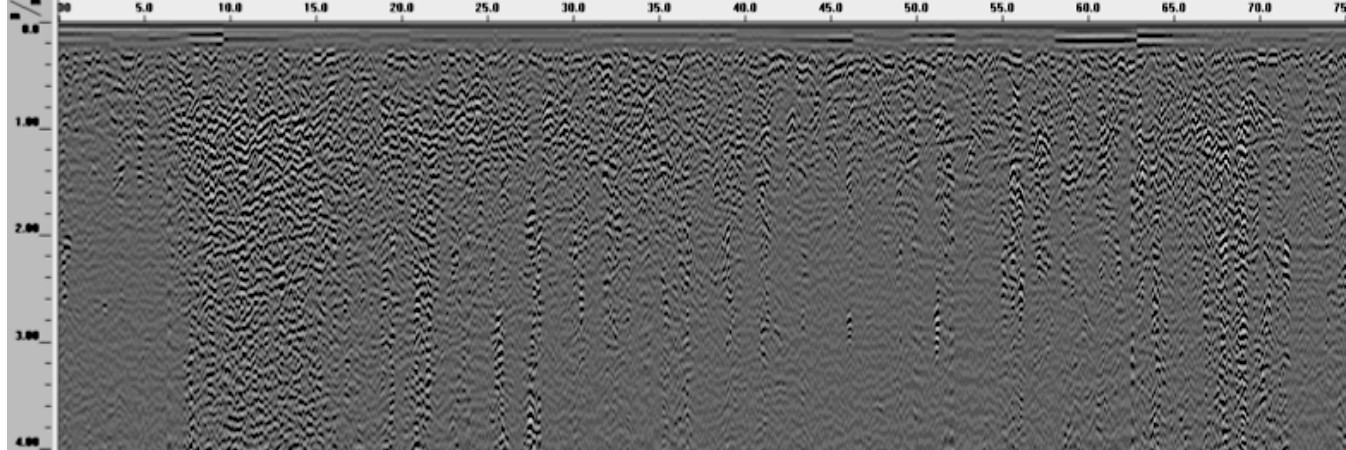

**Figure 5.** A sample 75 m section of profiles along transect S, without stacking or Hilbert transformation. Applying a wide (N=301 traces) background removal filter reduces antenna clutter but leaves diffractions and reflections unaffected. These raw data are dominated by columns of narrow diffractions, the widths of which show that they are caused by point targets. In neither the uncompressed (Figure 5) nor stacked (Figures 8 and 9) data for any profile are there hyperbolic diffractions sufficiently wide to be interpreted accurately for dielectric permittivity (Arcone, 1996; Yilmaz, 1987). The appearance of columns is discussed later.

The depth scale of the y-axes in Figures 6, 8, and 9 is based on a dielectric constant of 3, as justified below. The transverse profiles (Figure 8) are shown with a horizontal scale stacked by 10, while the much shorter Cross-PS and Cross-ZA profiles are not stacked (Figure 9). Profiles at 50 ns and 100 ns were recorded along each longitudinal transect. The elevation of the antenna when recording the profile over transect S is 8 cm greater than for other profiles. This places the deeper returns at higher gain in the transect S data.

All five Changri Nup profiles show absence of clear debris-ice interface reflections, energy concentrated in the near-surface followed by a significant signal decrease, and surface reflections partially interfered with by the DC. The volumetric returns from the near-surface indicate volumetric backscatter in the debris rather than reflections at distinct air-debris or debris-ice layer boundaries. The volumetric backscatter is dramatically stronger than the surface return, as would be expected for a rough, jagged surface. The radargrams in Figures 8, 9, B1, and B2 have been transformed into a positive instantaneous magnitude via an envelope function to indicate backscattered energy in the reflected waveforms (Figure 6). The Hilbert transform gives a smeared and streaked character to the backscatter in the radargrams. Figures B1 and B2 show that most signals return before 1 m depth.

Superimposed on the radargrams are absolute debris thicknesses denoted by *. Δ denotes thickness above frozen debris, which prevented digging to the glacier surface. The backscattered returns significantly vary in strength and delay, even over small lateral transect sections. The absence of a clear debris-ice transition in the radar data raised the question of whether the radar waves penetrated to the interface or were, instead, attenuated by scattering.



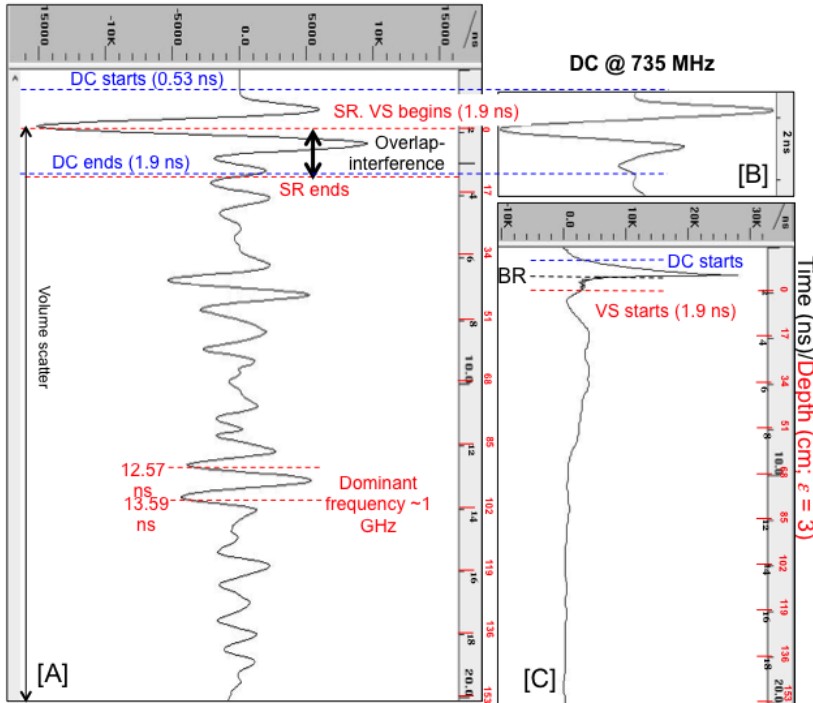

**Figure 6.** A trace from 170.6 m along Changri Nup Glacier, showing wave characteristics. Note that only the top 40% of the trace is shown. The blue box in Figure 8 shows the location of this trace. (a) The trace with no processing, corrected only for time zero. Labels and dashed lines of corresponding colors show durations of the direct coupling (DC), surface reflection (SR), and volume scatter (VS). The depth scale is based on $\epsilon = 3$, as derived from the experiments in this study. The DC and debris surface reflections overlap and, therefore, interfere. Note the lack of adjusted range gain greatly exaggerates signals at depths greater than ~30 cm. (b) Form and duration of the DC at the same time scale (i.e. y-axis) as in (a). (c) The trace from (a) with processing of: 10-fold stacking, background removal (BR), deconvolution, geometric spreading correction, and Hilbert magnitude transformation. The BR began at 1.27 ns and eliminated most subsequent superposition of the DC and surface reflection, the latter of which are mainly coherent (as revealed by experiments). Although volume scatter separates from the surface reflections at 3.4 ns, it technically starts at 1.9 ns. The range gain correction, which accounts for geometric spreading loss, has reduced trace amplitudes. Note that antenna unit height on the transect S profile was 27 cm. For the 19 cm antenna height used on the other 4 transects, the surface reflection lasts from 1.33 ns - 2.83 ns, and the volume scatter arrives at 1.33 ns.



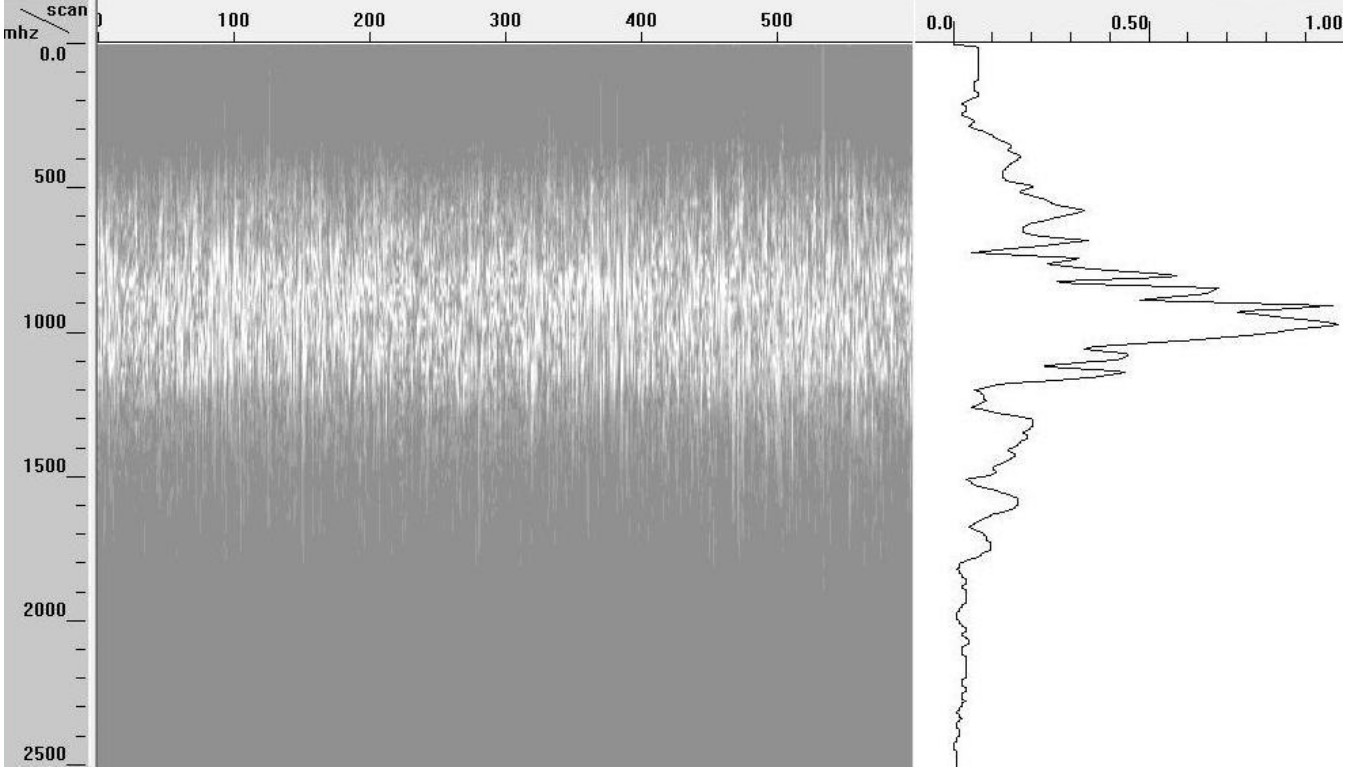

**Figure 7.** Reflected pulse spectra for transect S, before Hilbert transformation (with DC eliminated). For the large clasts (max size 18 cm, same as an *in situ* wavelength) 960 MHz dropped to about 800 MHz. For the smallest clasts it stayed near 1 GHz.

## 3.2   Rock Box Experiments

Our lab experiments investigated why clear glacier ice-debris interfaces were not recorded on Changri Nup Glacier. The experiments compared scattering behavior of pulses with two different spectra. They allowed us to determine the refractive index $n = \sqrt{\epsilon}$ and, thus, wave velocity through a dry debris-air matrix dominated by felsic clasts of three different size ranges.

5    We experimented with and without a layer of aluminum foil underlying the debris. The strong bottom reflection with foil showed that the 960 MHz pulse penetrated all three clast sizes with minor to no frequency loss or waveform dispersion (Figure 10) after a round trip in excess of 57 cm (approximately 3 *in situ* wavelengths) and that more penetration was possible. Table 2 shows that, regardless of clast size, the relative dielectric permittivity of the model debris was near 3. The received replication of the transmitted waveform shows little distortion despite the scatterers' being at least comparable to a pulse wavelength. In

10   fact, average reflected spectra varied from ∼ 770 MHz for the large clasts to 1 GHz for the small ones. This suggests that on Changri Nup most clasts were less than an *in situ* wavelength (18 cm) in maximum dimension.

In contrast to the 960 MHz signal, the 2.6 GHz one had so much scattering loss that there was no visible returned energy from the foil layer except beneath the smaller clasts. The 2.6 GHz GPR signal's difficulty penetrating medium or large clasts



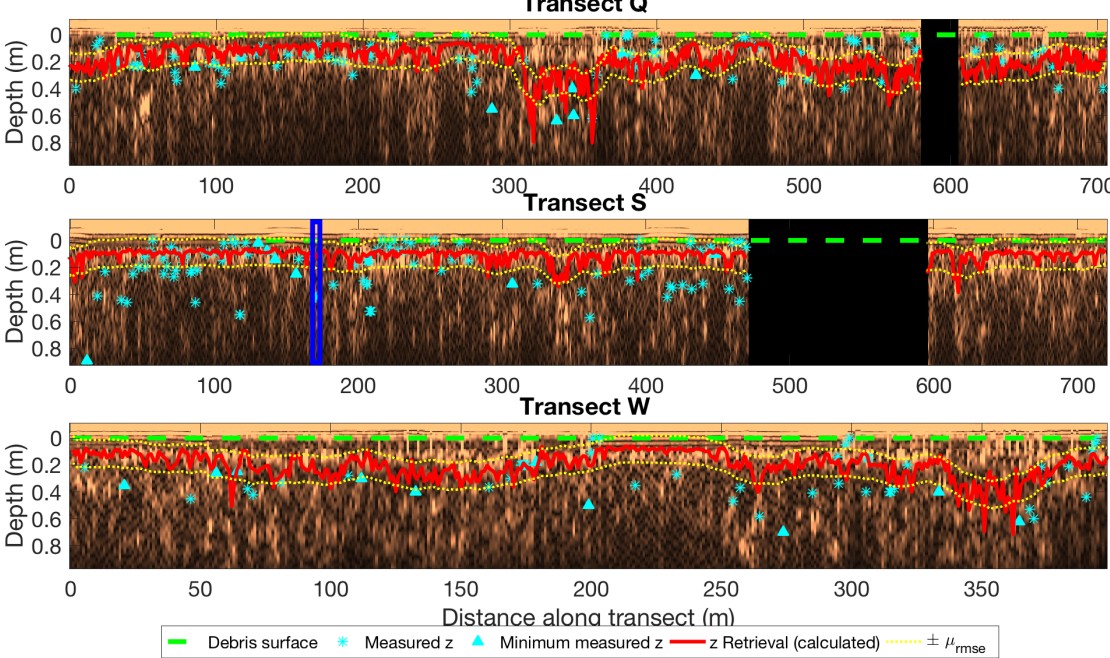

**Figure 8.** Hilbert-transformed GPR data, GPR-based thickness retrievals (red), and ground truth measurements along profiles over three transverse transects on Changri Nup Glacier. Uncertainty (yellow) is placed about a smoothed debris retrieval for ease of interpretation. All three run from West to East (climber's left to climber's right) across the glacier; the gap in Q indicates a corrupted file, and the gap in S is colocated with the prominent ice sail. The blue box in the profile taken along transect S corresponds to the trace in Figure 6. Experimental tests informed the location of the surface line (green) by allowing a calculation of the DC's 2 ns duration. The received signals are dominated by volumetric backscatter from the debris; subbottom returns arrive immediately after the surface reflection, which is weak and partly masked by the DC. Two types of ground truth measurements are shown; asterisks indicate excavations to the debris-ice interface, triangles indicate excavations that did not reach the interface and are thus a minimum debris depth. Radar depth scale was calculated using $\epsilon = 3$.

was likely because the medium debris compared in size to an *in situ* wavelength of 6.7 cm. It is likely that the 960 MHz signal would face difficulty penetrating sizes $\geq 18$ cm.

Also seen in Figure 10 is a faint surface reflection, coherent despite the rough surface and irregular dielectric structure between clasts and air. This coherence suggests that in the our field measurements, surface scatter was insignificant and mostly
5    removed by horizontal filtering.




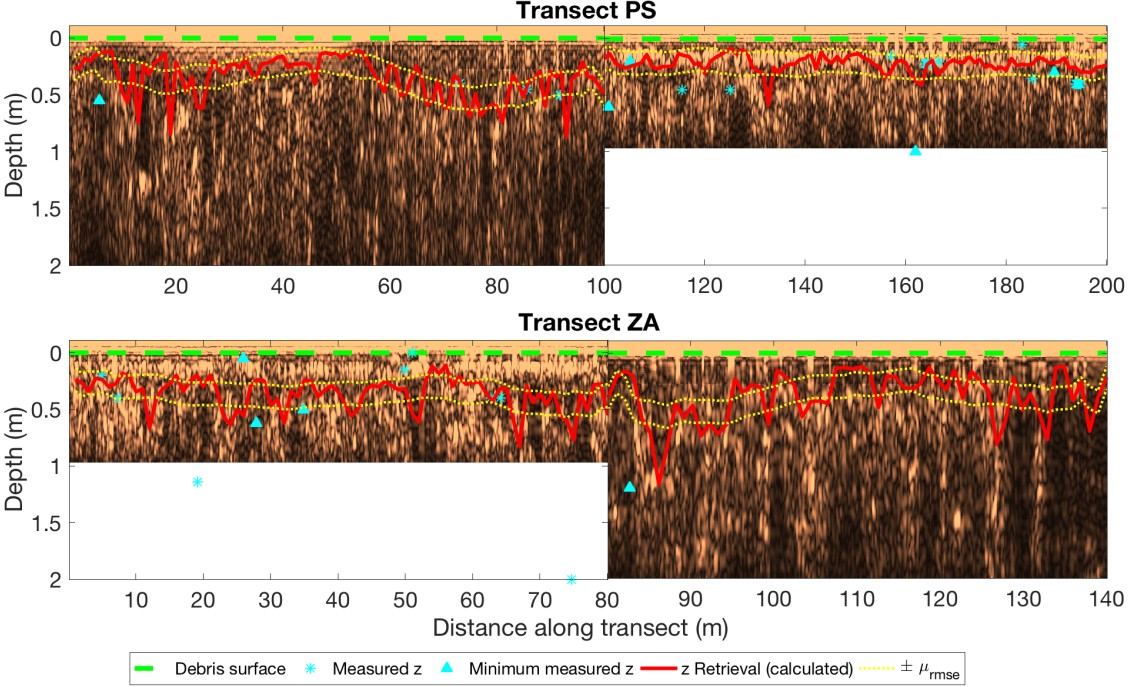

**Figure 9.** Hilbert-transformed GPR data, GPR-based thickness retrievals (red), and ground truth measurements along profiles over two longitudinal transects on Changri Nup Glacier, starting at their down glacier ends. Uncertainty (yellow) is placed about a smoothed debris retrieval for ease of interpretation. Signal characteristics are identical to what is described in Figure 8.

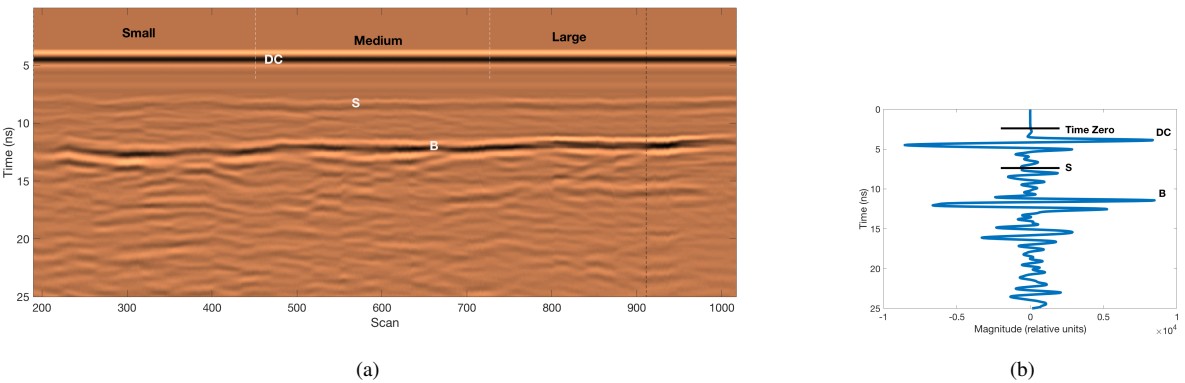

**Figure 10.** (a) Profile with raised 960 MHz antenna over rock box recorded May 8, 2018 with 28.6 cm (small), 29.6 cm (medium), and 33.6 cm (large) debris depths, with a single trace (b) from the large debris section indicated by the black line. Aluminum foil is below the rocks to provide a strong bottom reflection (B). The radargram (a) shows a surface reflection (S) that is smooth over all sections despite the rough surface, and the waveforms of the distinct backscattered events in the trace (b) replicate the transmitted waveform. The raised antenna allows separation of the S from the DC. Locations of S and B were determined from the known trough dimensions.



| Section | Small | Medium | Large |
|---|---|---|---|
| Debris depth (cm) | 28.6 | 29.6 | 33.6 |
| Refractive index ($n$) | 1.76 | 1.73 | 1.72 |
| Dielectric constant ($\epsilon$) | 3.08 | 2.99 | 2.97 |
| max error in $n$ (%) | 0.94 | 2.5 | 6.7 |

**Table 2.** Average refractive index ($n$) and dielectric constant ($\epsilon$) for six randomly chosen measurements in each size classification of the debris used in the experiment.

Profiles recorded without the aluminum foil permitted examination of the reflectivity at the debris-pine interface, which served as an analog to our field debris-ice interface. The average dielectric permittivity of the rock-air matrix (i.e. debris, $\epsilon = 3$) is comparable to that of ice ($\epsilon = 3.18$). Because the reflectivity of an interface is a function of the difference between the $\epsilon$ values of the two materials (see section 4.2), this similarity suggests that the reflectivity of the ice-debris interface was

weak and provides a reasonable explanation for the absence of a continuous ice horizon.

### 3.3 Thickness retrieval: Changri Nup

Experimental tests revealed that the DC lasts $\sim 2$ ns (shown in Figure 6), obfuscating any potential signal within that range. An antenna elevation in excess of 38 cm would have been necessary to achieve separation of the surface reflection and DC. Therefore, neither the 19 cm nor 27 cm heights used in the Changri Nup field tests was sufficient for complete separation of the

surface reflection from the DC, and the direct wave partially masked the surface reflection in all field data. Consequently, the surface reflection had to be located from its calculated return time rather than identified in the traces. The height of the antennas was sufficient, however, to produce a far-field spherical wave, the curvature of which approximated a plane when intersecting the surface, and our experiments show that surface scatter was weak and coherent.

For estimating the debris-ice interface on each trace, we used the area under the Hilbert transformed curve. Debris depth

measurements, excluding minima, from the main transverse transects informed the share of the area under the curve that indicated the debris thickness (see Figure 11 for visualization using an example trace of transect S). We used a leave-one-out cross validation test (LOOCV) with all measurements of debris surface – ice surface distance on Q, S, and W to determine a threshold $\tau$. LOOCV (Arlot et al., 2010) allows training of a model estimator (here, threshold $\tau$) with $n$ observations (here, depths). In each of $n$ iterations, the model is trained with all but one debris depth measurements, and the resulting estimates

are evaluated against the remaining observation. The procedure is repeated $n$ times, assuring generation of quality statistics.

Training the threshold (Figure 11) with LOOCV (Arlot et al., 2010) of depth measurements resulted in a $\tau$ of 38%. Thus, the depth by which 38% of the integrated energy over 1024 samples had been scattered matched most closely with the ground truth points. Upper integration bound $\tau$ is shown on an example Hilbert transformed trace in Figure 11 in black. The red curves in Figures 8 and 9, the thickness retrievals, are calculated with this $\tau = 38\%$. The yellow curves in Figures 8 and 9 indicate



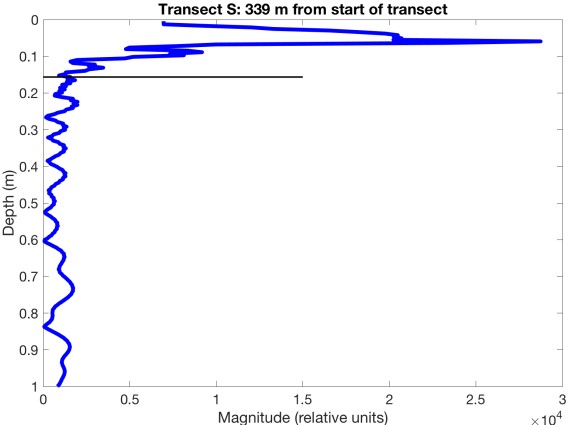

**Figure 11.** An example transformed trace from transect S. The upper bound of integration, which approximates the bottom of the debris layer, is shown as a black line. Only 256 samples are shown for clarity.

the uncertainty range, calculated from the mean RMSE (10 cm) of the LOOCV and subsequent smoothing to represent a 50 m moving average.

## 4   Discussion

### 4.1   Ground truth

5   Although there is broad-scale agreement between the calculated and measured average depths (Table 3), absence of an exact match is expected given that:

(a) Radar waves are transmitted in a finite, effective, two-way transmit-receive beamwidth ($70^o$), and consequently received signals represent an average of conditions proximal to the antenna. In contrast, the ground-truth depth measurements are point

10   measurements that are not necessarily representative of their surrounding areas.

(b) Due to practicality and field constraints, depth measurements are biased towards shallower porous debris and solid blocks, which are easier to measure than deep porous debris. Porous debris has $\epsilon = 3$ but solid granite $\epsilon = 5 - 7$ (Hubbard et al., 1997, field measurements of solid block). A wave travels more slowly in a material with a greater $\epsilon$. Using $\epsilon = 3$ expands

15   the depth scale relative to $\epsilon = 5$ such that measurements taken on solid rock used as training points in the LOOCV would bias the thickness retrieval toward shallow. Both (a) and (b) emphasize that we used a single value for the dielectric constant to calculate the depth scales on all five transects. Given debris' heteogeneous porosity and mineral composition, a single value is an average with some deviations to be expected.

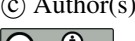



(c) Field measurements of debris cover are inherently difficult, and the mostly continuous dry, quartzo-feldspathic debris cover is punctuated by ice cliffs and local areas of wet debris, even in the post-monsoon season. Porosity ($\phi$) varies markedly over the 2 km of profiles. At the two locations where it was measured as 37%, the debris is relatively small and uniform. Over all five transects, however, it is generally larger and more variable than at either of the measurement locations; based upon
personal observations and photographs, $\phi = 50\%$ was used in this analysis.

## 4.2    Absence of interface

Our experiments suggest that the absence of a detectable debris-ice interface is due to a lack of dielectric contrast rather than loss of signal caused by scatter from the debris. We collected our profiles in the driest time of year, and mean temperatures
were well below freezing, so liquid water was highly unlikely to be present in the debris, and indeed was not found in any of our 353 ground-truth pits. Field observations during these ground truth excavations also suggest that bottom rock geometry is similar to surface geometry, and we find in the rock box experiment that the surface provides a smooth reflection (see Figure 10 and associated text). Therefore, the bottom of the debris layer should give a weak coherent reflection with most energy returned in the form of backscatter.
Dielectric contrast is the property of an interface between two materials that gives rise to a reflection, and the strength of that reflection is directly related to the strangth of the contrast. The reflection coefficient at the debris-glacier ice interface can be calculated from $\Gamma = \frac{n_{debris} - n_{ice}}{n_{debris} + n_{ice}} = 0.14$. A reflectivity this low would be difficult to detect, and thus the lack of a coherent reflection in our study, given the debris mineralogy and porosity, is not as surprising as it might seem.

The finding of $\epsilon = 3$ is reasonable for the mineralogy of Changri Nup's debris layer and the observed air content across all
five transects. As noted above, however, $\phi$ varies along transects. In areas of lower porosity, $\epsilon$ would be greater and the velocity less, which would compress the depth scale. Inversely, in more porous areas, the depth scale should be expanded relative to what is shown with $\epsilon = 3$. Estimating the effect of porosity using formulation of the Complex Refractive Index Method (CRIM, Equation 1, Arcone et al., 2016), whereby

$$\phi_{sd} * n_{sd} + (1 - \phi_{sd}) * n_{air} = n_{debris}, \tag{1}$$

is rearranged to

$$\frac{n_{debris} - (1 - \phi_{sd}) * n_{air}}{\phi_{sd}} = n_{sd} \tag{2}$$

where $n_{air} = 1$ and $\phi_{sd} = 50\%$, gives a refractive index for solid debris ($n_{sd}$) of 2.46, thus relative permittivity $\epsilon_{sd}$ of 6.07. Therefore, a debris sample that EM waves pass through with a 1.4 ns two-way travel time is 25 cm thick with $\phi_{sd} = 50\%$ but 21 cm and 30 cm with $\phi_{sd} = 70\%$ and $\phi_{sd} = 30\%$, respectively.
A value of 6.07 for $\epsilon_{sd}$ is within the range of most solid feldspars, 5.9-7 (Hubbard et al., 1997). A change in rocktype to limestone (with $\epsilon_{calcite-dolomite} = 9$), the plausible end member in the Khumbu Himalaya, would give $\epsilon_{debris} = (.5 * \sqrt{(9)} +$





$.5 * (1))^2 = 4$. However, less porous debris of either rocktype would give a vastly different dielectric: with 30% porosity, for example, $\epsilon_{Changri\ Nup} = 4.1$ and $\epsilon_{limestone-rich\ debris} = 5.7$. Both give a greater dielectric contrast with glacier ice ($\epsilon = 3.18$). A dielectric constant of 4.03 on the debris cover of Koxkar glacier (Huang et al., 2017) supports the idea that debris covers vary in their dielectric properties across HMA.

These calculations may illustrate a difference between our study and others. McCarthy et al. (2017) detected a debris-ice interface on 16/29 profiles, meaning that nearly half of their recorded profiles lacked an interface. In contrast with our conclusions, they attributed this lack to debris too thin, debris too thick, and high scatter. Eleven of their profiles showing an interface were recorded with frequencies lower than 900 MHz, which would decrease scatter but fail for thin layers. Nicholson and Mertes (2017) and Nicholson et al. (2018) profiled Ngozumpa glacier at 200 and 600 MHz and identified a debris-ice
interface throughout their data. None of these studies give a detailed description of debris mineralogy or porosity, and it is certainly plausible that variations from ours in either or both would account for a dielectric contrast at the glacier surface.

### 4.3   Backscatter

The scatter dominating our records – both from the field and from the rock box – is volumetric. The rock box results demonstrate a surface reflection (Figure 10), showing that the later returns are not wide angle surface scatter. Further, multiple volumetric
scattering appears insignificant because our time delays give reasonable and consistent values for dielectric constant in the rock box (Table 2), and the backscattered waveforms in both the field and rock box replicate the transmitted pulse. Therefore, the events are from single scattering.

Volumetric point backscatter (i.e. simple hyperbolic diffractions) from jagged debris accounts for the general appearance of the profiles before stacking and Hilbert transform. The shapes of the surface debris represent underlying debris, while the
sizes are generally larger on the surface. Streaks in the transformed profile indicate energy returned from multiple depths at a single location. These streaks are likely due to the combination of diffractions from the edges of angular debris and the Hilbert enveloping of each pulse. Resonance within an occasional flat rock could also explain streaks but seems unlikely because multiple reflections within a layer rapidly lose strength. Some returns may have been caused by ice-embedded debris, which may explain many events that occur beneath thickness retrievals.

Englacial debris is more concentrated in the longitudinal profiles (Figure 9) than the transverse ones (Figure 8). In the accumulation zone of Changri Nup, there is a large rocky spur generating many rockfalls. Debris eroding from this single, dramatic spur reemerges on the surface as a thick ridge in the glacier's midline, to the west of the salient ice sail in Figure 1. Both longitudinal profiles cover this ridge, and Figure 9 shows substantially more deep returns than Figure 8.

Although deeper returns and resonance features exist in our data, experiments showed that most of the signal's return energy
is volume scattering from within the debris layer.

Huang et al. (2017) interpret debris thickness from the volume scattering power of polarimetric L-band (1.27 GHz) SAR. They assume that volume scattering in the ice is negligible. They used a mathematical derivation of a volume scattering – thickness relation from a measured coherency matrix, and then used target decomposition to determine scattering sources (surface, volume, double-bounce). This approach using a broadband pulse worked only for debris thicknesses less than 50 cm.



Our approach differs in that we do not have a thickness limitation. Our short-pulse GPR experiments allowed us to identify the volume and surface scatter directly.

## 4.4 Leave-one-out cross validation

The LOOCV suggested that the time by which 38% of the integrated energy has been scattered corresponds to the depth of the debris layer. The mean depth of the retrieval is greater than the mean depth of field measurements for transect Q but smaller than that of all other transects. Still, the measured averages are within the retrieval error range for transects Q, S, and W. The error bounds are large because of the variation in depths measured in a medium that varies horizontally on the cm scale. The measured average thicknesses for the longitudinal transects exceed the retrieval's upper error bound, which is unsurprising given the dearth of ground truth points and measurement bias towards thick, exposed blocks. The two longitudinal transects are much shorter than transverse transects and cover the thick debris band on Changri Nup Glacier.

Evaluating integration bounds for each transverse transect separately rather than together (38%) yields 35%, 43%, and 42% for transects Q, S, and W, respectively. These thresholds give mean retrieval depths of $15 \pm 10$ cm, $17 \pm 9$ cm, and $26 \pm 8$ cm. The retrieval thicknesses calculated per transect do not vary significantly from those calculated with the aggregation of all transverse profile measurements. Therefore, the upper integration bound for using volumetric backscatter as a proxy for debris thickness is likely not location-dependent on a single glacier or glacier-dependent unless there is a significant difference in debris composition.

| Transect | Number of ground truth points | Length (m) [measured] | Mean thickness[1] (and $\sigma$) from ground truth (cm) | Mean thickness (and $\sigma$) from thickness retrieval (cm) | $\tau$ |
|---|---|---|---|---|---|
| Q | 116 | 706 [681] | 17 (24) | $19(10) \pm 10$ | 35% |
| S | 132 | 720 [595] | 16 (20) | $11(5) \pm 10$ | 43% |
| W | 65 | 398 | 25 (14) | $21(10) \pm 10$ | 42% |
| Cross-ZA | 8 | 140 | 59 (66) | $37(18) \pm 10$ | – |
| Cross-PS | 32 | 200 | 35 (12) | $29(13) \pm 10$ | – |
| All transects | 353 | 2014 | 23 (23) | 19 (9) | 38% |

**Table 3.** Ground truth measurements and thickness retrievals for each transect. $\tau$, the threshold value determined by LOOCV, is 38%. The uncertainty range is given by the mean RMSE (10 cm) of the difference in measured - retrieval depths: 10 cm is the mean of a range (1.2E-2 - 49 cm) that is large because of the high variation in measured thicknesses. RMSE of the debris retrieval calculations is 1.2E-3. (1) Mean thickness excludes measurements of minimum thickness.

## 4.5 Glacier-scale considerations

Given mechanisms of debris transport, input distribution, and higher temperatures at lower elevations (Scherler et al., 2011), it is expected that debris thickness increases downglacier, from the clean ice-debris transition to the terminus. The means of



the measurements and the thickness retrieval indicate that transect W has debris that is thicker than that on upglacier transects Q and S, which are closely spaced. However, the mean thickness on S exceeds that on Q, demonstrating local variability contrasting with the glacier-wide trend. In fact, local-scale variability exists everywhere in the debris cover (Nicholson et al., 2018). Variability at the transect scale is caused not only by surface features such as ice cliffs, rivers, and ponds but also by the

debris source location. We expect thickness to be greater in the glacier flowlines from rockfall-prone areas and to be less in the flowlines of large clean zones of the accumulation area.

In future work, it would be useful to explore variability of debris thickness by source of the debris (which ridge), comparing thickness distribution on different glacier flowlines. A first-order look at the lateral variability along our profiles reveals that Q and S do, indeed, have thicker debris along the ridge sourced from the eroding rock spur. Excluding this ridge from the mean

thickness along these transects does not give significantly different statistics, however, because many of the depth measurements on the ridge from Q ($\sim 290 - 470$ m) were minima, and shallow debris measurements dominate the mean depth on S over the ridge ($\sim 250 - 410$ m) although they are not spatially representative. Future work is needed comparing flowlines to regularly spaced depth measurements, without the biases described in Section 4.1. Extrapolation and interpolation of thickness data on debris covered glaciers is an important area of future research.

**5  Conclusions**

Our study calculated debris thickness along five transects of Changri Nup Glacier based upon the depth decay of the volumetric backscatter that dominates our recorded profiles. We pursued this approach because we did not detect a reflection from the ice surface; due to the dielectric properties (close to that of ice) which arise from its mineralogy and porosity, Changri Nup's debris cover lacks dielectric contrast with glacier ice. We explored how GPR signals interact with a debris cover of angular,

felsic debris; namely, we showed 960 MHz penetration through irregular rocky debris, volumetric backscatter and weak surface scatter despite debris sizes comparable to an *in situ* wavelength, and a dielectric permittivity ($\epsilon = 3$) relatively constant over a range of clast sizes. Future research examining the variability in dielectric across a debris layer is necessary for quantifying the impact of our assumption of a constant $\epsilon$.

Our aim was to explore the efficacy of potential remote systems operating near 1GHz, since they engender more efficient

measurement of debris thickness on the glacier scale. Our findings suggest that systems operating near this frequency could successfully estimate dry debris cover thicknesses on large scales based on depth of volumetric backscatter. Our main challenges were navigating rough terrain in the field and analyzing data from a field system that needed greater elevation to separate surface scatter from the DC. Relative to satellite systems, airborne ones are more feasible for sounding radar because of the trade-off between range and bandwidth. Radiating a GPR pulse requires a short antenna, a consequence of which is a wide

beamwidth. Therefore, such a system would be deployed most feasibly on low-flying drones.

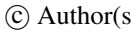


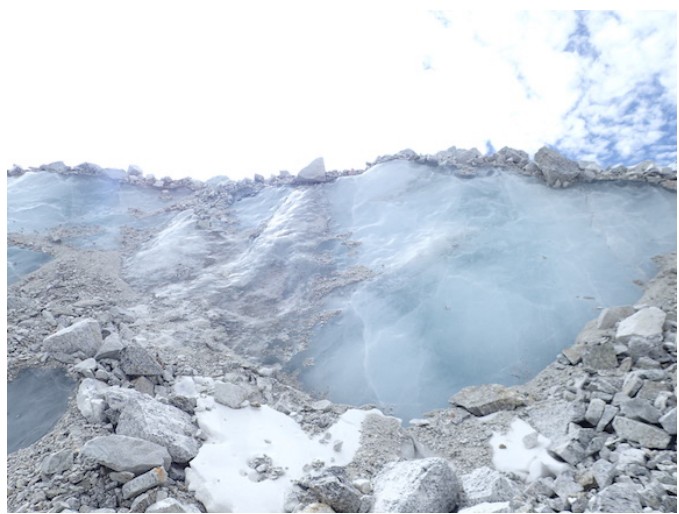

**Figure A1.** Photograph 1/2 of transect Q. Note thin debris cover over mostly clear ice. Some parts of transects went along cliffs such as this one, others went up and over similar features.

*Code and data availability.* By the time this manuscript review is complete we will post raw radar data files and ground-truth data to a durable repository (hopefully with NSIDC but possibly elsewhere). Along with the data, we will post the processing code used for this work to go from raw data, through the LOOCV sequence, to the final figures for the manuscript. The code will likely be posted on github, and will be crosslinked to the data. If a repository exists that will allow posting of the code and data in one place, we will use that.

5 **Appendix A: Photos by transect.**

**A1   Transect Q**

**A2   Transect S**

**A3   Transect W**

**A4   Transect Cross-PS**

10 **A5   Transect Cross-ZA**

**Appendix B: All collected data**

**B1**

*Author contributions.*





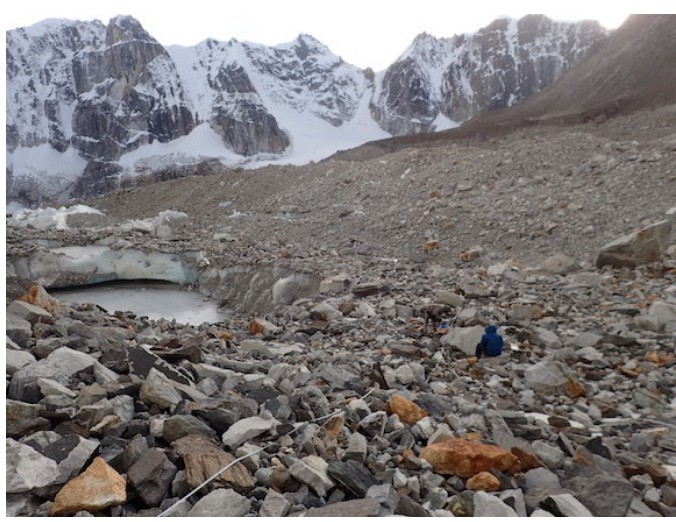

**Figure A2.** Photograph 2/2 of transect Q. Exposed ice is visible to the left, note field workers for scale in middle right.

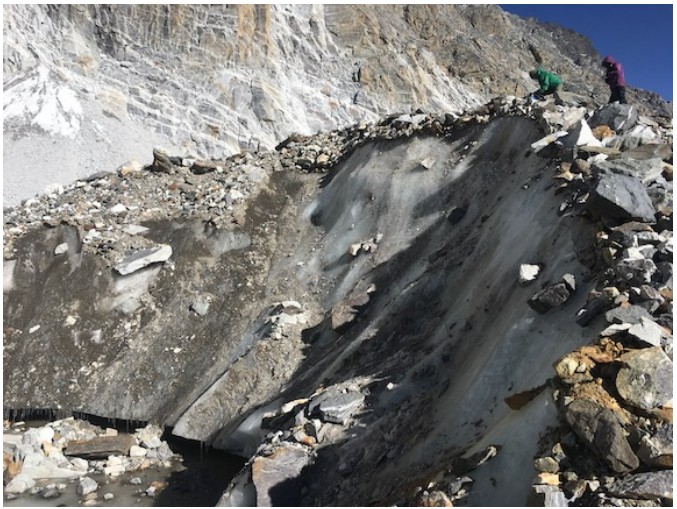

**Figure A3.** Photograph 1/3 of transect S.





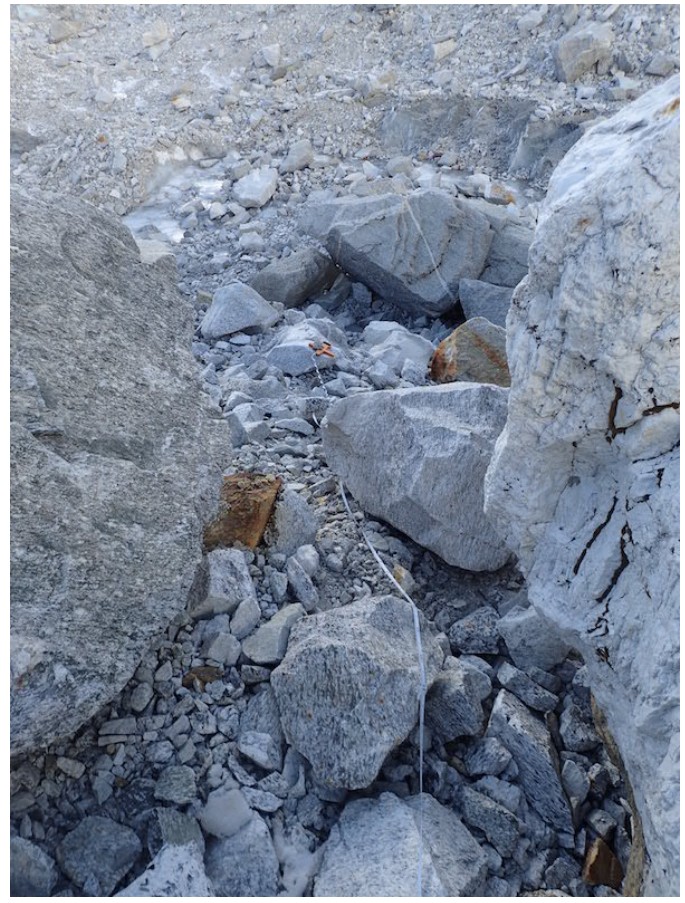

**Figure A4.** Photograph 2/3 of transect S. Note significant clast size variability.

*Competing interests.* The authors declare that no competing interests are present.

*Acknowledgements.* Thanks go to the Ev-K2-CNR Project and Nepal Academy of Science and Technology for fieldwork logistics. Funding for an initial, exploratory field season (2014) came from Dr. Summer Rupper and the Brigham Young University, with additional funds from the American Alpine Club. Dr. Mike Dorais and Josh Mauer (BYU) provided field assistance to the first author. Funding for the work presented in this paper (2015) came from the Department of Earth Sciences at Dartmouth College, the American Philosophical Society, and the Albert Cass Travel Fellowship. Gabriel Lewis, Sonam Futi Sherpa, and Dr. Dibas Shrestha assisted Alexandra Giese with data collection. Ian Raphael assisted the US-based authors with laboratory test set up. Feedback from Dr. Jonathan Chipman improved this manuscript.





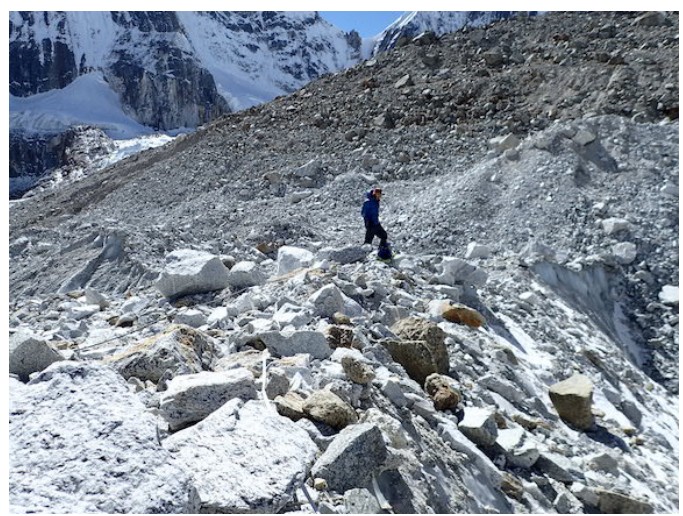

**Figure A5.** Photograph 3/3 of transect S. In spite of the large-scale roughness the debris cover continued to be relatively shallow.

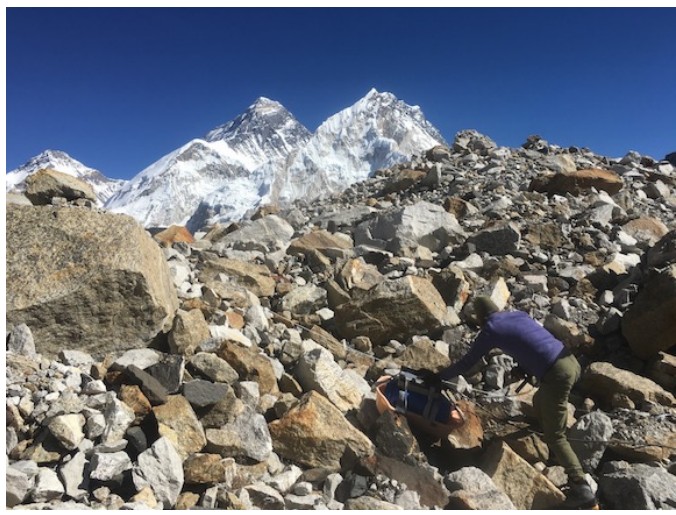

**Figure A6.** Photograph 1/2 of transect W. Note extreme terrain.





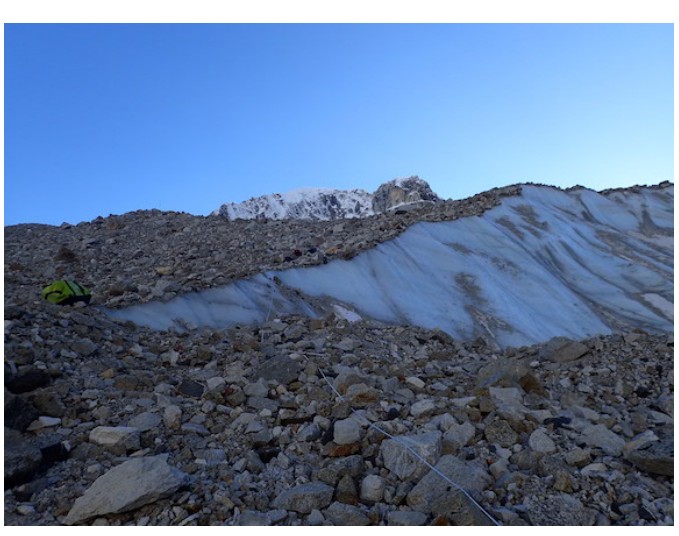

**Figure A7.** Photograph 2/2 of transect W. Note duffel bag for scale at left.

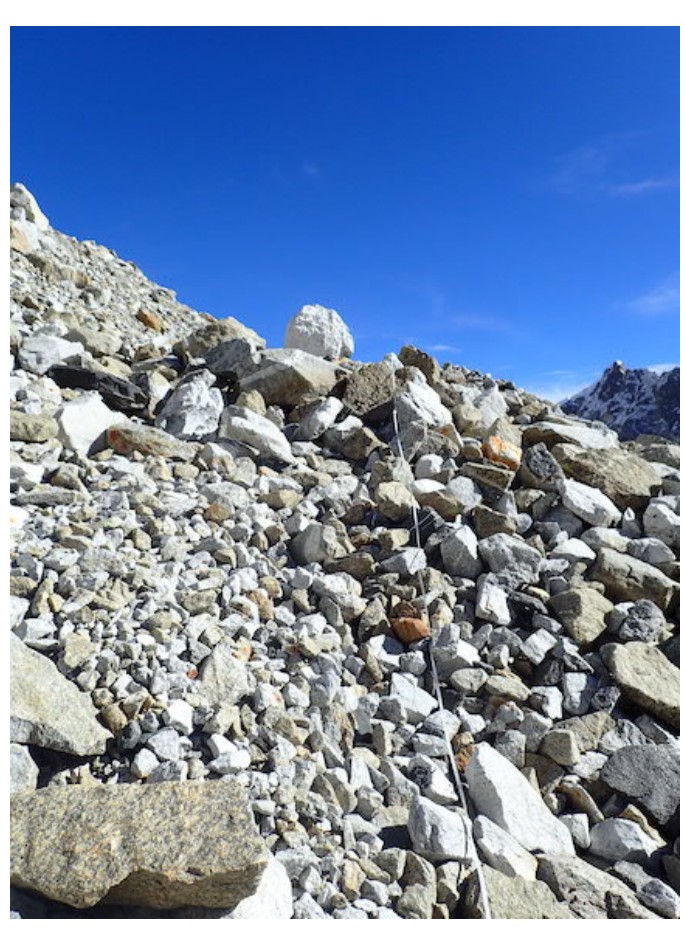

**Figure A8.** Photograph 1/2 of transect Cross-PS.





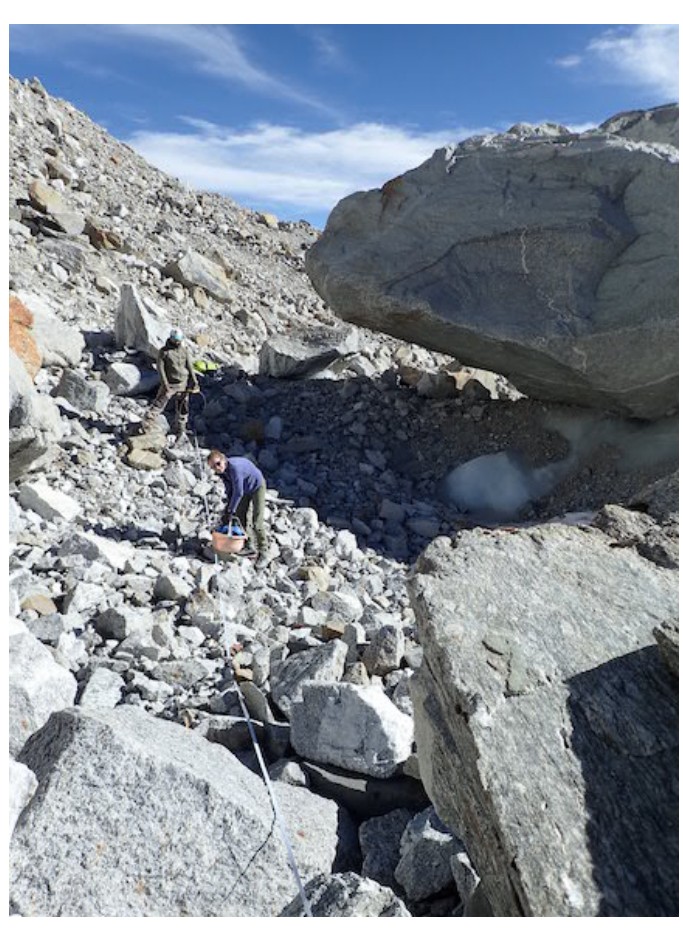

**Figure A9.** Photograph 2/2 of transect Cross-PS.

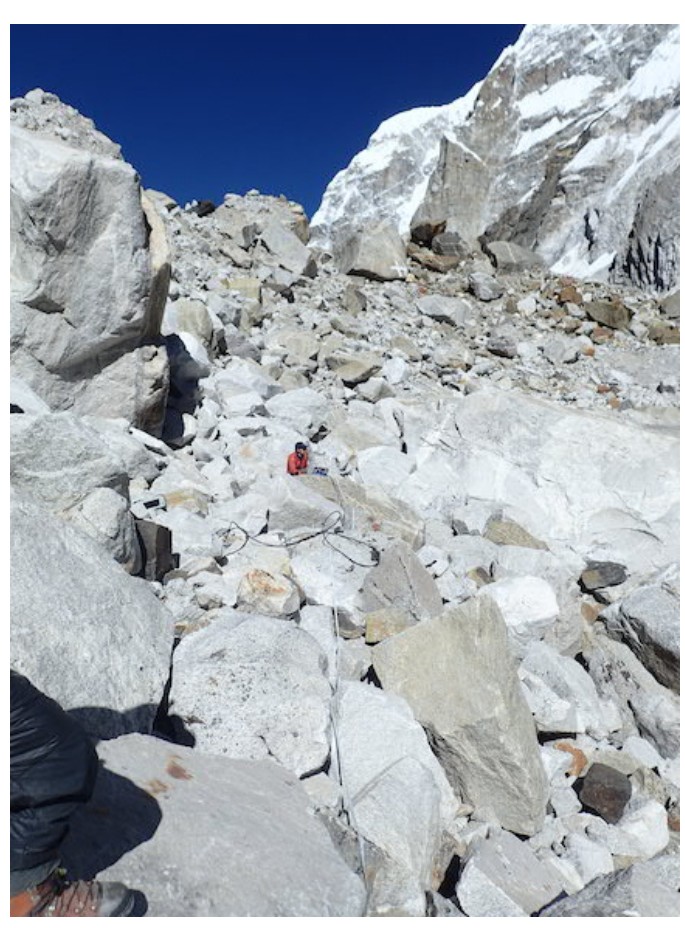

**Figure A10.** Photograph 1/2 of transect Cross-ZA. Note field worker for scale in center.



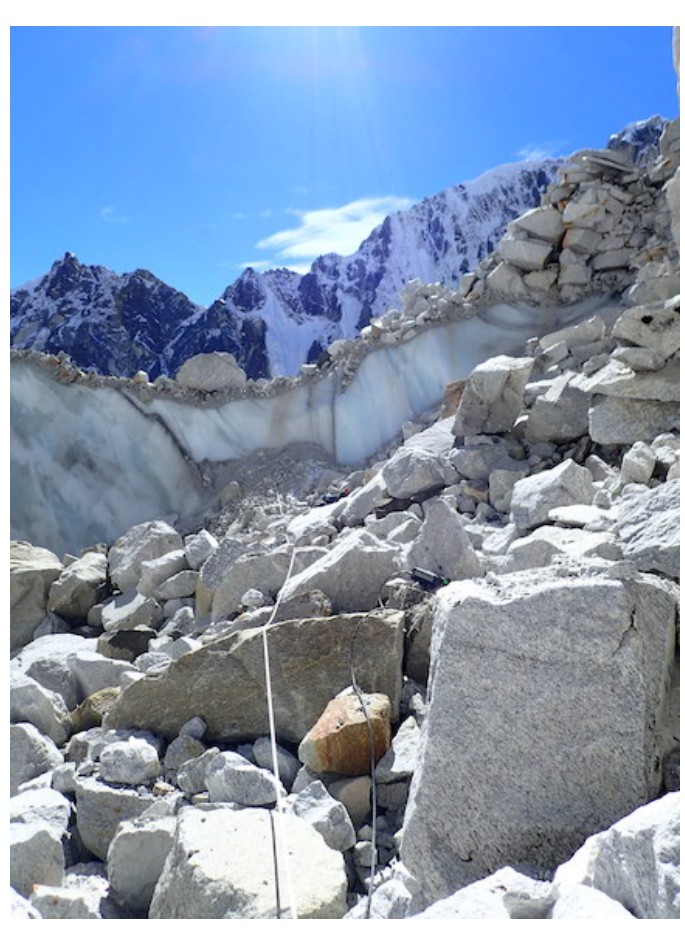

**Figure A11.** Photograph 2/2 of transect Cross-ZA.




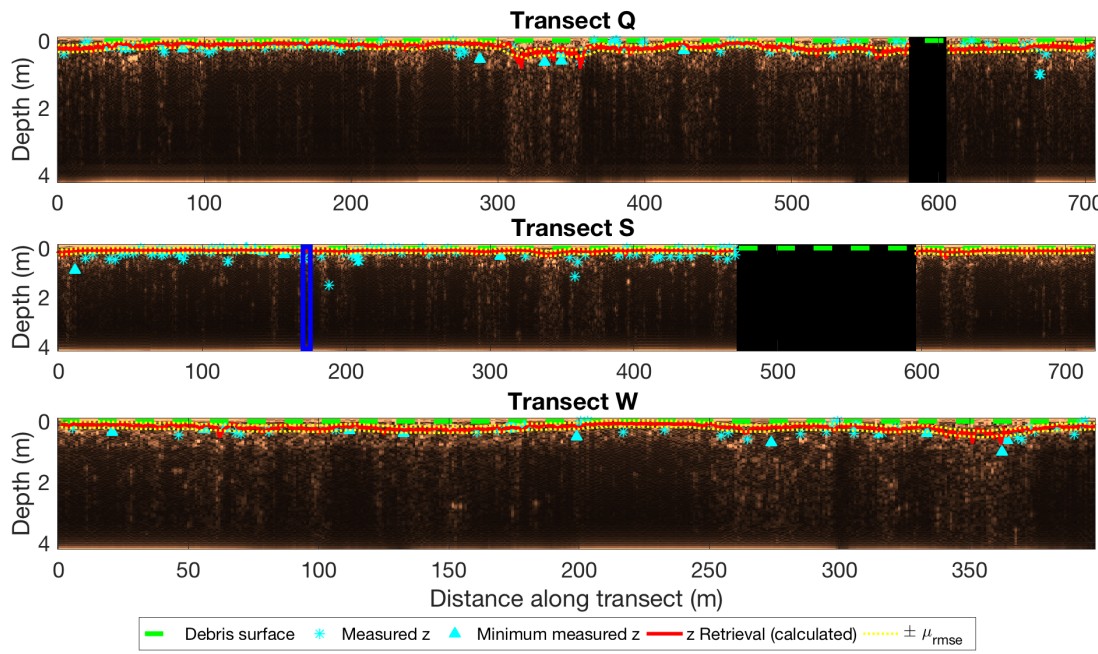

**Figure B1.** Hilbert-transformed GPR data, GPR-based thickness retrievals (red), and ground truth measurements along profiles over three transverse transects on Changri Nup Glacier. Uncertainty (yellow) is placed about a smoothed debris retrieval for ease of interpretation. All three run from West to East (climber's left to climber's right) across the glacier; the gap in Q indicates a corrupted file, and the gap in S is colocated with the prominent ice sail. These figures are equivalent to those in Figure 8 but show all 1024 samples instead of only the near-surface.



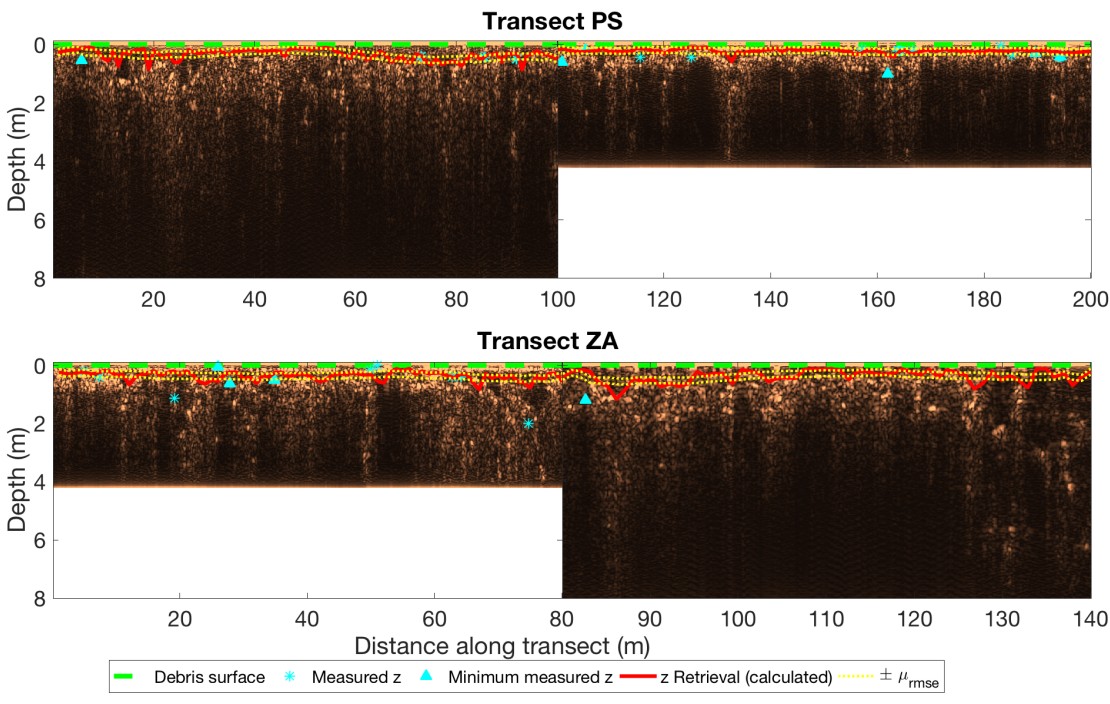

**Figure B2.** Hilbert-transformed GPR data, GPR-based thickness retrievals (red), and ground truth measurements along profiles over two longitudinal transects on Changri Nup Glacier, starting at their down glacier ends. Uncertainty (yellow) is placed about a smoothed debris retrieval for ease of interpretation. These figures are equivalent to those in Figure 9 but show all 1024 samples instead of only the near-surface.



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
