# Peer review of "The response of supraglacial debris to elevated, high frequency GPR: Volumetric scatter and interfacial dielectric contrasts interpreted from field and experimental studies"

_The Cryosphere, 2019_

## Referee Comment (RC1) · Michael McCarthy (Referee) · 23 Jun 2019

General comments: This paper asks whether 'airborne' ground-penetrating radar (GPR) can be used as a means of quantifying supraglacial debris thickness. The authors collected a large amount of GPR data on a debris-covered glacier in Nepal, as well as on a sort of artificial debris-covered glacier in a 'laboratory'. In addition, they made extensive manual pit measurements of debris thickness on the glacier. They found no clear ice surface reflection below debris from which to estimate debris thickness in the field-based GPR data and suggest that this is because the ice and debris

had similar dielectric constants during the survey period. However, they found that they could estimate debris thickness from volumetric backscatter in the GPR data with some success compared to the manual measurements, using a statistical approach, instead.

The research question is a good one because debris thickness is a strong control on the surface energy and mass balance of debris-covered glaciers, and because making ground-based measurements is very difficult. The data collection will have been hard work, having been carried out at high altitude and in a remote part of Nepal, and the dataset that was collected is potentially a valuable resource for the modelling community. The analysis of the dataset, which was possibly more difficult than the authors had originally hoped it would be, is honest, thorough and appropriate. The work is novel in the sense that it focusses on the potential for airborne application (achieved by mounting the GPR on a plastic box above the debris surface), and in its development of the above-mentioned statistical approach to determining debris thickness from volumetric backscatter. The development of the volumetric backscatter method is a useful one, and is the primary contribution this paper makes to the field, as well as a good starting point for any planned or future studies of debris thickness by airborne GPR. It seems likely that the method would always have to be used in conjunction with some ground-based measurements, for calibration, which is a downside and, although 'estimated' debris thicknesses are broadly similar to 'measured' debris thicknesses, I expect that if the two were plotted against each other there would be a lot of scatter. However, the idea is still attractive, especially given the recent proliferation of UAV-use in glacier research, and the difficult nature of the problem. The paper is generally well-written and easy to read, although a few things could be clarified and some of the figures need axis labels and clearer numbering.

I have three major issues that should be addressed before publication (as follows), as well as a number of more specific comments and some technical corrections.

(1) The dielectric constant that was used to estimate debris thickness in the field was determined through laboratory experiments. Even though the laboratory debris was

chosen to have broadly similar mineralogy, grain size etc. to the field debris, the two will inevitably have had slightly different dielectric properties. This is an important point because the choice of dielectric constant will have affected the threshold derived in the volumetric backscatter method, and therefore the estimated debris thicknesses. I don't think the analysis needs to be done again, because optimising the threshold against the pit measurements will have compensated for such a difference, but this should be stated more clearly in section 3.3. (2) I am not convinced that the pine shavings used in the lab are a good analogue for ice. The dielectric contrast is considerably greater between the debris and the pine than between the debris and the ice. I suppose the point is that if there is no clear reflection at the interface between the debris and the pine, where there is a relatively strong dielectric contrast, it should be expected that there is no clear reflection between debris and ice, where there is a relatively weak contrast? If this is the case, I think this could be explained more clearly. (3) On p18, in Table 3, the standard deviations are larger than the mean, suggesting a non-normal distribution if debris thickness is always positive. Median debris thicknesses and interquartile range might be more appropriate here.

Specific comments: P1, L2. The first paragraph of the introduction could do with restructuring. The first sentence is about water supply, the second, third and fourth about debris-covered glaciers, the fifth back to water supply, then the sixth to climate change. Maybe the water supply and climate change sentences could go together instead.

P3, L10. I don't see why the frequency affects the area that can be covered using GPR.

P3, L12. Remove 'implying a layer of solid granite or very dense debris'. Any number of sediment or debris layers with a dielectric constant of 6.46 are possible. Therefore a dielectric constant of 6.46 does not imply a layer of solid granite or very dense debris.

P4, Figure 1. Would be good to have the transects labelled and the caption updated accordingly.

P5, Table 1 caption. Why did you add 1 cm? This seems quite arbitrary. Please explain

and include the explanation in section 2.2 main text, rather than in table caption.

P7, L26. Sentence beginning 'We collected...' seems like it would fit better in section 2.2 as it is more about data collection than results.

P9, Figure 5. Needs axis labels and clearer numbers.

P10, Figure 6. Needs axis labels and clearer numbers.

P11, Figure 7. Needs axis labels and clearer numbers.

P16, L3. This is the first mention of the porosity measurements. Should be mentioned in section 2.2 as well.

P17, L5. McCarthy et al (2017) discussed the possibility that the strength of the ice surface reflection in their GPR data was variable due to variable dielectric contrast. It seems likely that they could often see an ice surface reflection in their GPR data because the ice was often melting during their study period. I would suggest that this is the key difference between the two studies.

P17, L25. Suggest 'Assuming the small reflections in the profiles are englacial debris, englacial debris is more concentrated...'. There is no proof that these reflections are englacial debris, so some caution is required.

P18, L1. There must be a thickness limitation unless the debris cover is completely lossless. Possibly it is better to say there is no thickness limitation within the range of debris thicknesses that is likely to be encountered in a supraglacial setting.

P19, L12. I am not sure of the meaning of the sentence beginning 'Future work...'. This should be clarified.

P29, Figure B1. Y-axis limits are too wide to see any detail near the debris-ice interface. Suggest 0 – 4 m would be sufficient.

P30, Figure B2. Y-axis limits are too wide to see any detail near the debris-ice interface.

Suggest 0 – 4 m would be sufficient.

Technical corrections: P2, L26. Sentence beginning 'Because debris thickness...' needs to be rearranged, otherwise 'its thickness' – 'it' meaning 'debris thickness' – reads 'debris thickness thickness'.

P3, L21. Remove 'yet'.

P23, L31. Suggest 'to the southeast'. 'Southeasterly' suggests flow from the southeast, as e.g. westerlies are winds blowing from the west.

P4, L9. Suggest 'longitudinal to ice-flow direction'.

P4, L12. By 'collecting more thickness data', do you mean making pit measurements? Please clarify.

P7, L29. 'Spectra' is plural, yet we have 'is shown in Figure 7'.

P10, Figure 6. Text on figure suggests volume scatter and direct coupling both begin at 1.9 ns, yet these two lines are separated by at least a nanosecond in time. Possibly one is a typo?

P11, L3. 'compared the scatter behaviour'.

P16, L5. Remove 'personal'.

---

## Referee Comment (RC2) · Anonymous Referee #2 · 18 Sep 2019

The response of supraglacial debris to elevated, high frequency GPR: Volumetric scatter and interfacial dielectric contrasts interpreted from field and experimental studies

Alexandra Giese, Steven Arcone, Robert Hawley, Gabriel Lewis, and Patrick Wagnon
Review for The Cryosphere

This is a novel and timely piece of work developing techniques for quantifying debris thickness on debris-covered glaciers from high frequency GPR using the characteristics of volume scattering of radar returns where a distinct reflection from a debris-ice
boundary is absent. As the authors state, there is a growing interest in mapping debris cover from space, from the air and from the ground because of its impact on the energy balance and melt processes. The paper develops methods for analysing relatively high frequency radar data collected from just above the debris layer (so without direct contact with the surface) over a glacier in the Nepal Himalaya (960 MHz) and over an artificial laboratory debris layer covering pine board (960 MHz and 2.6 GHz) in terms of the attenuation of volume backscatter, and using this to calculate debris thickness. Errors are discussed.

General Comments A slight difficulty I had with the paper is it is a hybrid of a very technical GPR methods paper and a paper which applies those methods to mapping debris thickness across a glacier and then interprets them in terms of glacial processes. This is reflected in the title of the paper, which is not attractive and would not encourage many people to read it. It is also reflected in the overall structure of the paper, where field data are analysed and presented, using information that is dependent on the results of the laboratory work, which is not presented until later. And it is also reflected in the aim of the paper which is very brief (and then is followed by a summary of what was done) and stated on Page 3 lines 20-22: "Our aim was to find how a frequency relevant to remote radar systems (i.e. $\sim$ 1 GHz) performs in glacial debris. To this end, we compared the depth of volumetric backscatter from GPR data with ground truth measurements of debris thickness. We validate our indirect backscatter method with experimental studies".

I would encourage the authors: to think about the aims of the paper and to better articulate those following a relevant literature review; to think whether the overall structure of the paper could be improved, perhaps putting the experimental lab work before the presentation of the field work, if the former is needed to interpret the latter; and then to come up with a better, less convoluted, more snappy, more engaging title for the paper.

A key element to the paper is the recognition that there is no obvious reflection from the ice at the base of a relatively thin debris layer. It is concluded (with apparent supporting evidence from the laboratory work) that this is not because of a lack of coupling between the radar and the surface but because the permittivity of the debris on Changri Nup Glacier is essentially the same as the underlying ice and so there is no dielectric contrast. This contrasts with the results from other (admittedly few) similar studies. I found the presentation of results, analysis, discussion and logic of the argument behind this point very difficult to follow as it seemed particularly labored and spread over many parts of the paper in different sections. Is it possible to present the results and the analysis in a much more focused and coherent way so that the reasons for a lack of reflection between debris and ice is more convincing? Is it really nothing to do with the lack of coupling between radar and the debris? Is it to do with the radar frequencies used? Is there really something special about the debris and ice on this glacier compared to glaciers studied in previous work? Do the authors have any advice on what radar frequencies should be used in the future? Or whether different radar frequencies should be used in different settings? Or whether the time of year and the presence / absence of water would have any bearing on the results? I found the argument concerning the lack of reflection particularly difficult to follow partly because of the terminology regarding "dielectric" and "permittivity". Could the authors ensure that these terms are being used correctly and consistently throughout the paper? For example: P1. L11 dielectric contrast P3 L14 relative dielectric permittivity P9 L1 dielectric constant P12 L3 irregular dielectric structure P14 L2 dielectric permittivity P17 L14 dielectric P17 L17 dielectric properties

Are some of these actually the same thing? On P6 L1-2 it is stated that a 960 MHz antenna transmits a pulse with dominant wavelengths of $\sim$ 31 cm in air and $\sim$ 18 cm in debris with a relative permittivity of 3. So it is assumed a priori that the debris on the on the glacier has a relative permittivity of 3 (is this the same as a dielectric constant?) and this is subsequently justified with reference to the artificial lab experiments.

Detailed comments P1 L8-11. These two sentences do not quite seem logical to me and read like a circular argument/ tautology. Can you clarify precisely what the lab

results show, and what this means in terms of the interpretation of field data? P1 L18-20. Something not right here. Debris covers 14-18% of glacier area in Himalaya. But then you say it's even greater in the East at 25%. Is the 14-18% figure wrong? Or confined to the West and Central Himalaya? P2 L 4-5. This sentence seems to belong with sentences on lines 19-22 so suggest delete from here and move to below or vice versa. P2 L5 say "be used to measure" instead of "indicate"? P2 L6. What about other sources of debris on glaciers, e.g. what about extra debris from mass movements from valley sides / lateral moraines? And possibility, at least, of subglacial debris brought to surface by thrusting? P2 L 33. Say 200 and 600 to be consistent with above. P2 L33. Say "...Glacier, Nepal..." P2 L35 – P3 L 6. This reads like results / discussion. Suggest move from here to later. Exception could be point about inability to drag antennas. Why is use of low frequencies "irrelevant"? Why is frequency relevant or not for areal coverage? P3 L14-16. Something not right about this sentence. Also it presents results again. We need only a priori justifications for the methods. Or some reference to "preliminary data collection and analysis" or some such. P3 L16. Should hypothesized be present tense? P3 L17. What do you mean by "favorable"? For what? Do you mean "high"? P3 L18-19. This seems to be the key a priori reason for your approach. Is it the case that clasts are large at Changri Nup and dragging antenna is difficult, therefore you need a different approach involving use of backscatter information? I think you need a better articulation of the limitations of previous work, the differences between CN glacier cf. other glaciers studied, and therefore a better justification for your work culminating in a clear set of aims. P4. Fig 1 etc. the term "cross" to refer to the transects running up and down glacier seems odd to me. I think of "cross glacier" and "cross transects" as going across a glacier and "along glacier" or "longitudinal transects" as going up and down glacier. P5. Last sentence of Fig Heading. Is what is stated really obvious from the photos? Doesn't appear so to me. P6. L5. Consider "Although, as shown later, this raising caused..." P7. L10. Why is N & M 2017 referenced here? P7. L14. "local" to what? P7. L25. This sentence doesn't make sense to me. P7. L28. Suggest mention things here in text that are currently in the

Fig Heading. Also, important to explain the "surface reflection" is debris surface not ice surface. P8 Fig 4 Caption. Change "Schematic" to "Annotated photograph"? P9. Poor text quality in Fig 5. Y axis. In Fig caption suggest change sentence to something like: "In none of the stacked or uncompressed data profiles are the hyperbolic diffractions wide enough to be used to calculate the dielectric permittivity." P 9. L6. Could delete the glacier name here. Also, is this sentence correct? The profiles show an absence of 3 things? Seems odd. Wouldn't it be best to state what they do show? P9 L9. Suggest replace "dramatically" with "much". P9 L16 "raises" P11. L9. Delete apostrophe after 'scatterers'. P12. Fig 8 Heading. Need consistency in referring to colours in key. Either refer to them all or none. Delete "climber's left to climber's right". Suggest move penultimate sentence to higher up to discuss all the lines in the Fig. first. P13 Fig 9. Explain the reason for the gaps in the traces. Same with Figure B2 in Appendix. P13. Fig 10 Heading and Fig 6 Heading earlier. Need to better explain what "surface reflection" is. You have "surface reflection" but also "debris surface reflection". Surface reflection is abbreviated SR and S inconsistently. Why is bottom reflection not defined in Fig 6? P14 L 1. "pine-board" ? P14. L2-4. How do these sentences follow on from the previous sentence? I don't understand what the evidence for this is. P14 L10-12. What is the evidence for this? Refer to a Figure and describe / label the Figure? P14 L13. "To estimate. . ." P14 L14-15. What does this mean? Are you just using the measurements to calibrate an equation to determine depth from radar data? P14 L 15. The procedure needs to be explained wrt Fig 11. P14 L17. Threshold tao needs explaining. You're referring to a threshold in the % of an area under a curve right? P14 L17-18. Confusion here. What is an iteration? If it's the number of times the "model" is run then it's run n-1 times right because you leave one out each time? P14 L19. ". . .assuring generation of quality statistics". Poor English. P14 L 22. Can't Figs 11 and 12 be combined? Could use colors to show 20, 30, 40, etc % of area under curve, then show a line for 38%. P15 L5-6. Suggest "There is broad-scale agreement between the calculated and measured average depths (Table 3). An exact match is not expected because:. . ." P16 L3. Delete "field measurements of solid block)" I assume? P16 L 5. Suggest ". . .shallow depths.

[Figure]

This, together with point (a) above, emphasize..." P16 L11. Are you referring to debris grain size here? Small and uniform? P17. L3. "would" not "should" to be consistent with L2. P17 L20. "...lack to debris too thin, debris too thick, and high scatter" Poor English. Rewrite. P17 L21. "...but fail for thin layers". What does this mean? P18 L5&6. No need for new paragraph here as you're discussing the same point. P18 L30. These numbers here are different to those in Table 3. Why only refer to the 3 cross glacier transects rather than the other 2 here? P18 L31. It is really the case that these thicknesses "do not vary significantly"? P20 L7. ..."dielectric permittivity across..." P21-32. Appendixes have got mixed up with the references. Might be useful to show location and look direction of all the photos on a Map (e.g. Fig 1).
* * *

---

## Referee Comment (RC3) · Anonymous Referee #3 · 24 Sep 2019

Summary: This study employs new lab experiments into understanding GPR signal properties for sounding glacier debris cover, and presents a new approach to analysing GPR backscatter strength from field surveys as a proxy for debris thickness. I think the lab experiments are particularly useful and I'd like to see them expanded to more thoroughly explore the backscatter signal. I am less convinced by the derivation of debris thickness from the summed backscatter power, as explained below. As such, I suggest further work is needed, and recommend that this manuscript not be accepted but a new manuscript be submitted at a later date.

Details: Page 3 Line 10: "...low frequencies irrelevant for efficient areal coverage..." There is no reason why the systems used in the cited studies (and much larger systems) could not be deployed by helicopter (I have a helicopter-slung system more than 20 m long for example). "...dragged antennas, an approach that was impossible at our field site..." The surface of Ngozumpa, Lirung and Langtang glaciers are very similar to the pictures of Changri Nup shown here.

Line 26: "frequency relevant to remote systems..." by which you mean drones. See above comment regarding helicopters, which are readily available for charter in the Khumbu. Page 4 Line 11: The frozen ice surface is critical here. By far the largest control on the dielectric contrast in such a setting is the presence or absence of water. It would be much easier to detect the ice-debris interface if you surveyed in dry weather by in thawing conditions (so debris is largely dry but ice surface is wet). This is what McCarthy et al. and Nicholson et al. did.

Page 9 Line 3: "Profiles at 50 ns and 100 ns..." Not sure what this means, please clarify. Line 8 and in general: I don't think the term 'volume scatter' is used correctly. It is normally used to describe the net result of multiple bounces from many discrete, closely-spaced reflectors of similar size to the wavelength, within some medium, which I think is what you have here. In such a case, the interfaces are likely to be air-rock, rock-rock and ice-rock interfaces. I think you're describing it as single-bounce reflections back from individual point scatterers within the debris layer, which is analogous to detecting an aircraft in flight (which clearly is not volume scatter). If it was dominated by single-point scatterers then you'd expect to see hyperbolae in the unstacked radargram (e.g., Figure 5) resulting from the radar moving closer to then further away from the point scatterer, but I don't see these. It's also not clear why the signal would penetrate through the debris above but reflect from these particular buried scatterers. Line 9: "The volumetric backscatter is dramatically stronger than the surface return..." But the surface return is obscured/interfered with by the direct wave, so it is not clear what the surface return strength is from the field data. Indeed, the lab experiments

show the surface return at least as strong as the volume scatter (Figure 10).

Page 11 Rock box experiments: these are really nice and I'd like to see more, with the aim of characterising the attenuation and ice-surface detectability. I suggest: Increasing the debris thickness (round trip of 57 cm is thickness of only 28.5 cm, which doesn't capture the potential range of thicknesses in the field). Vary the thickness and measure how the aluminium-base signal strength varies, to characterise attenuation with depth. With the pine base, try wetting the pine (by pouring water in at the base, while keeping the debris largely dry) to simulate a thawing ice surface. Try the above but wetting the debris from above, to simulate rain or snowmelt. Repeat the above with a range of GPR frequencies.

Page 14 Line 14: "...area under the Hilbert transformed curve..." – meaning that the value calculated is the sum of the gain-corrected backscatter power over the chosen time window? This would seem to depend on both the gain function used and on the time-window selected. As the signal attenuates with depth, eventually it reaches the noise floor (i.e., system noise), so if a long time window was chosen for exactly the same profiles, this would add 'power' to the summed magnitude because the this approach would sum up this noise. This would increase the total summed power and so the 38% level would be deeper. This would be a spurious increase, unrelated to the actual debris thickness. Also, I think this approach assumes a linear relationship between summed backscatter power and debris thickness, but the effect of attenuation would mean that this is incorrect. This is because the near-surface returns contribute much more to the sum than the deeper ones, so for example, doubling the debris thickness would not double the summed backscatter power, the increase would be much less than double. I think the sensitivity of the summed power to increasing thickness would follow a decay curve. Clearly the deeper debris should contribute less backscatter because much of the transmitted signal has already been scattered back by the debris above, and so is no longer available. This also means that the variation in the threshold calculated locally (e.g., the 35%, 42%, 43% in Table 3) could mean large variability in

thickness, i.e. including another 8% of the summed power could mean including a lot more debris, because the debris at depth contributes little to the total. A further implication of attenuation is that there is certainly a limit to the debris thickness that can be quantified by GPR (page 18, line 1, "...we do not have a thickness limitation.") – above a certain thickness, there is no longer enough signal for backscatter to be detected, and therefore no sensitivity to debris. It is not obvious a priori what this limit is because it depends on the debris properties, but once this is exceeded than all that can be said is that the debris is thicker than this detectability limit. Finally, the 38% threshold is empirical and so does depend on the local properties (of porosity, lithology, grain size distribution, wetness). This means that it may be useful for interpolating thicknesses within a single survey, but is unlikely to be universally applicable (as suggested later).

Line 24: If the debris layer produces 38% of the backscatter, where does the other 62% of backscatter come from? Seems that you've already ruled out significant backscatter from the surface and sub-debris ice.

Page 16 Line 13 and elsewhere: be careful with the use of the word 'coherence' – it has a particular meaning in radar processing, whereas I think you're using it to mean 'detectable and continuous' or similar. Line 29: need to add these porosity uncertainties into the debris-thickness error budget (yellow lines in Figure 8 etc). Page 17 Line 11: by far the greatest influence on ice detectability in at least the McCarthy and Nicholson studies is the wet ice surface. Section 4.3: see above regarding volume scatter versus point scatter.

---

## Author Comment (AC1) · 21 Oct 2019

Big changes: (1) The dielectric constant that was used to estimate debris thickness in the field was determined through laboratory experiments. Even though the laboratory debris was chosen to have broadly similar mineralogy, grain size etc. to the field debris, the two will inevitably have had slightly different dielectric properties. This is an important point because the choice of dielectric constant will have affected the threshold derived in the volumetric backscatter method, and therefore the estimated debris thicknesses. I don't think the analysis needs to be done again, because optimising the

threshold against the pit measurements will have compensated for such a difference, but this should be stated more clearly in section 3.3.

We have incorporated the suggestion to make this explicit in Section 3.3 ("Thickness retrieval: Changri Nup") and additionally discussed it in Section 4.5 ("Uncertainty.")

(2) I am not convinced that the pine shavings used in the lab are a good analogue for ice. The dielectric contrast is considerably greater between the debris and the pine than between the debris and the ice. I suppose the point is that if there is no clear reflection at the interface between the debris and the pine, where there is a relatively strong dielectric contrast, it should be expected that there is no clear reflection between debris and ice, where there is a relatively weak contrast? If this is the case, I think this could be explained more clearly.

Indeed, reflectivity for debris-ice = 0.015 is an order of magnitude less than reflectivity for debris-pine = 0.11 (given in Sections 2.3, along with added justification for choosing the set-up). The calculation of and explanation for reflectivity is given in the Methods, and the relative magnitude is discussed subsequently. The intention was to construct an interface whose reflectivity (0.11) was on the order of (i.e. very low) but greater than that on the glacier to investigate wave behavior at an interface; although the magnitudes differ, they are both far below a reflectivity that would be detected under the debris medium we studied. A greater reflectivity in our constructed setup allowed us to observe processes that occurred during data collection on Changri Nup but which were not detected (in a way, enhancing them for the purpose of our investigation). We also improved the Discussion section description of this.

(3) On p18, in Table 3, the standard deviations are larger than the mean, suggesting a non- normal distribution if debris thickness is always positive. Median debris thicknesses and interquartile range might be more appropriate here.

Median debris thicknesses and interquartile ranges have been added to the table and the observation that standard deviations are larger than the mean added to the Discussion text.

Specific comments:
P1, L2. The first paragraph of the introduction could do with restructuring. The first sentence is about water supply, the second, third and fourth about debris-covered glaciers, the fifth back to water supply, then the sixth to climate change. Maybe the water supply and climate change sentences could go together instead.

We rewrote and restructured the Introduction; now, the initial sentence is no longer about water supply, and the introduction does not jump between topics.

Small changes (line numbers refer to TCD version):
P2, L26. Technical corrections: Sentence beginning 'Because debris thickness. . .' needs to be rearranged, otherwise 'its thickness' – 'it' meaning 'debris thickness' – reads 'debris thickness thickness'.

This sentence was eliminated in the revised manuscript.

P3, L10. I don't see why the frequency affects the area that can be covered using GPR.

We removed this to avoid confusion. The thought behind this statement was that a lower frequency GPR system could not, practically, be run from a drone (which would have vast areal coverage).

P3, L12. Remove 'implying a layer of solid granite or very dense debris'. Any number of sediment or debris layers with a dielectric constant of 6.46 are possible. Therefore a dielectric constant of 6.46 does not imply a layer of solid granite or very dense debris.

Removed.

P3, L21. Remove 'yet'.

Removed.

P3, L31. Suggest 'to the southeast'. 'Southeasterly' suggests flow from the southeast,

as e.g. westerlies are winds blowing from the west.

Fixed.

P4, Figure 1. Would be good to have the transects labelled and the caption updated accordingly.

Both have been done.

P4, L9. Suggest 'longitudinal to ice-flow direction'.

Language changed to across- and along-glacier in response to Reviewer 3.

P4, L12. By 'collecting more thickness data', do you mean making pit measurements? Please clarify.

By this statement, we meant collecting transect A. The language has been revised to clarify/specify.

P5, Table 1 caption. Why did you add 1 cm? This seems quite arbitrary. Please explain and include the explanation in section 2.2 main text, rather than in table caption.

We added 1 cm when calculating the hypotenuse distance of a triangle with geometry established by the antenna manufacturer. As this detail is not important to our methods or results and seemed to confuse the reader, we removed it.

P7, L26. Sentence beginning 'We collected. . .' seems like it would fit better in section 2.2 as it is more about data collection than results.

This sentence was, indeed, more about data collection than results. We removed it from the Results section.

P7, L29. 'Spectra' is plural, yet we have 'is shown in Figure 7'.

Fixed, and this material is now in the Supplementary Materials.

P9, Figure 5. Needs axis labels and clearer numbers.

This figure was completely remade and has axis labels and clearer numbers.

P10, Figure 6. Needs axis labels and clearer numbers.

This figure was updated in the Supplementary Material P10, Figure 6. Text on figure suggests volume scatter and direct coupling both begin at 1.9 ns, yet these two lines are separated by at least a nanosecond in time. Possibly one is a typo?

This was, indeed, a mistake in the original manuscript and has been updated in the new figure.

P11, Figure 7. Needs axis labels and clearer numbers.

This figure was updated in the Supplementary Material

P11, L3. 'compared the scatter behaviour'.

This sentence in the original submission is not in the revised manuscript.

P16, L3. This is the first mention of the porosity measurements. Should be mentioned in section 2.2 as well.

The porosity measurements that we collected (to inform a constant in a moisture diffusion model) are not relevant to the porosity in the Complex Refractive Index Method calculation for GPR and have been removed from the manuscript.

P17, L5. McCarthy et al (2017) discussed the possibility that the strength of the ice surface reflection in their GPR data was variable due to variable dielectric contrast. It seems likely that they could often see an ice surface reflection in their GPR data because the ice was often melting during their study period. I would suggest that this is the key difference between the two studies.

We expanded the Discussion to include season, reflecting reviewer comments, in the "Absence of interface" section.

P16, L5. Remove 'personal'.

Fixed.

P17, L25. Suggest 'Assuming the small reflections in the profiles are englacial debris, englacial debris is more concentrated. . .'. There is no proof that these reflections are englacial debris, so some caution is required.

Fixed; the suggested text is included in the new manuscript.

P18, L1. There must be a thickness limitation unless the debris cover is completely lossless. Possibly it is better to say there is no thickness limitation within the range of debris thicknesses that is likely to be encountered in a supraglacial setting.

Fixed; the suggested text is included in the new manuscript.

P19, L12. I am not sure of the meaning of the sentence beginning 'Future work. . .'. This should be clarified.

This has been rewritten and clarified.

P29, Figure B1. Y-axis limits are too wide to see any detail near the debris-ice interface. Suggest 0 – 4 m would be sufficient.

P30, Figure B2. Y-axis limits are too wide to see any detail near the debris-ice interface. Suggest 0 – 4 m would be sufficient.

The point of these figures was to illustrate the entire time range of our data. Although these figures have not been changed, the point of showing all collected data has been made more clear in the text: we wanted to show, by including data from the entire time range used, that most of the energy is returned from the near-surface.
* * *

---

## Author Comment (AC2) · 21 Oct 2019

General Comments: A slight difficulty I had with the paper is it is a hybrid of a very technical GPR methods paper and a paper which applies those methods to mapping debris thickness across a glacier and then interprets them in terms of glacial processes.

We restructured the paper in response to this feedback; the most significant change was ensuring the main text of the manuscript was written for a glaciology readership and moving the technical details about GRP to the Supplementary Material for the

interested reader.

- This is reflected in the title of the paper, which is not attractive and would not encourage many people to read it. - It is also reflected in the overall structure of the paper, where field data are analysed and presented, using information that is dependent on the results of the laboratory work, which is not presented until later. - And it is also reflected in the aim of the paper which is very brief (and then is followed by a summary of what was done) and stated on Page 3 lines 20-22: "Our aim was to find how a frequency relevant to remote radar systems (i.e. âĹij 1 GHz) performs in glacial debris. To this end, we compared the depth of volumetric backscatter from GPR data with ground truth measurements of debris thickness. We validate our indirect backscatter method with experimental studies".

I would encourage the authors: (1) to think about the aims of the paper and to better articulate those following a relevant literature review;

We revised the Introduction section and carefully defined and articulated the scope and aims of the study. We feel that the literature review was sufficient and that there are simply limited references; we cite literature of GPR on debris covered glaciers and, additionally, related studies outside this niche (e.g. rounded "box-of-boulders experiment," glacial till, debris covered glacier studies in the Antarctic dry valleys)

(2) to think whether the overall structure of the paper could be improved, perhaps putting the experimental lab work before the presentation of the field work, if the former is needed to interpret the latter;

We restructured the paper considerably from the initial submission to ensure that Methods, Results, and Discussion points were in their appropriate sections. Furthermore, we incorporated the reviewer's observation that some of the arguments were based upon the results of the rock box experiment and presented before the experiment was described. We maintain that the field data are the most important data of our study and, therefore, keep the discussion of the rock box after that of the field data. However, we revised the abstract and introduced significant modifications to the manuscript text to emphasize the role of the rock box experiments: to understand the data from Changri Nup Glacier.

(3) and then to come up with a better, less convoluted, more snappy, more engaging title for the paper.

We have renamed the paper, "Measuring supraglacial debris from elevated, high frequency GPR."

A key element to the paper is the recognition that there is no obvious reflection from the ice at the base of a relatively thin debris layer. It is concluded (with apparent supporting evidence from the laboratory work) that this is not because of a lack of coupling between the radar and the surface but because the permittivity of the debris on Changri Nup Glacier is essentially the same as the underlying ice and so there is no dielectric contrast. This contrasts with the results from other (admittedly few) similar studies. I found the presentation of results, analysis, discussion and logic of the argument behind this point very difficult to follow as it seemed particularly labored and spread over many parts of the paper in different sections. Is it possible to present the results and the analysis in a much more focused and coherent way so that the reasons for a lack of reflection between debris and ice is more convincing?

Yes, and we have restructured and significantly reworked the new manuscript. Additionally, we expanded the "Absence of interface" section in the Discussion with more in-depth explanations. We also moved that section to be first in the discussion to feature it for readers.

Is it really nothing to do with the lack of coupling between radar and the debris? Is it to do with the radar frequencies used?

The rock box experiments showed that the coupling is sufficient; there is enough energy reaching the interface to be detected if it were detectable (i.e. that the energy is getting

to the interface is evident from the strong reflection from the aluminum foil in the rock box). At the higher frequency we tested in the rock box, energy did not reach the interface through all clast sizes, but the rock box results clearly demonstrated that the 960 MHz system we used on Changri Nup could have detected an interface of sufficient dielectric contrast.

Is there really something special about the debris and ice on this glacier compared to glaciers studied in previous work?

It is difficult to say conclusively, but the main difference may have been liquid water. In the "Absence of interface" section, we discuss how the timing of our study differed from others' studies, which may have detected an interface due to the presence of liquid water at the ice-debris interface during the melt season (the time of data collection).

Do the authors have any advice on what radar frequencies should be used in the future? Or whether different radar frequencies should be used in different settings?

Because we did not set out to test a range of frequencies, we feel that such advice is beyond the scope of this manuscript. The aim of our study was to investigate how GPR operating your 1 GHz performs in supraglacial debris. We investigated a bandwidth centered near 1 GHz for several reasons: first, lower frequencies may not resolve a complex interface between debris and glacier ice. Second, we knew from an initial, exploratory field season that average thickness was roughly only 30 cm. At a lower frequency and this thickness, we risked having interference between the surface and bottom reflection. While it is true that, at a lower frequency such as used by others, the scattering might be less and the power radiated stronger, we were interested in exploring a frequency that could be deployed over a range of debris thicknesses

Or whether the time of year and the presence / absence of water would have any bearing on the results?

This is now addressed in the updated "Absence of interface" section, which includes a

discussion of the season of data collection.

I found the argument concerning the lack of reflection particularly difficult to follow partly because of the terminology regarding "dielectric" and "permittivity". Could the authors ensure that these terms are being used correctly and consistently throughout the paper? For example: P1. L11 dielectric contrast P3 L14 relative dielectric permittivity P9 L1 dielectric constant P12 L3 irregular dielectric structure P14 L2 dielectric permittivity P17 L14 dielectric P17 L17 dielectric properties Are some of these actually the same thing?

The updated "Absence of interface" section presents the arguments explaining the lack of reflection more clearly, logically, and comprehensively than in the original manuscript. Additionally, we now include definitions of dielectric and permittivity and state explicitly that they are used interchangeably in the manuscript.

On P6 L1-2 it is stated that a 960 MHz antenna transmits a pulse with dominant wavelengths of âĹij 31 cm in air and âĹij 18 cm in debris with a relative permittivity of 3. So it is assumed a priori that the debris on the on the glacier has a relative permittivity of 3 (is this the same as a dielectric constant?) and this is subsequently justified with reference to the artificial lab experiments.

All instances of a priori assumptions based on the later rock box experiments have been removed and/or reordered in the rewrite of the manuscript.

Detailed comments (line numbers refer to initially submitted manuscript, not the version prepared for TCD)

P1 L8-11. These two sentences do not quite seem logical to me and read like a circular argument/ tautology. Can you clarify precisely what the lab results show, and what this means in terms of the interpretation of field data?

Yes. The new Abstract does clarify what the lab results showed and how they aided in the interpretation of field data: "The laboratory tests suggest that the ice-debris

interface return was missing in field data because of a weak dielectric contrast between solid ice and porous dry debris."

P1 L18- 20. Something not right here. Debris covers 14-18% of glacier area in Himalaya. But then you say it's even greater in the East at 25%. Is the 14-18% figure wrong? Or confined to the West and Central Himalaya?

The 14 – 18% figure is a regional average; the eastern part has a debris coverage greater than the average. This has been rephrased for clarity.

P2 L 4-5. This sentence seems to belong with sentences on lines 19-22 so suggest delete from here and move to below or vice versa.

The Introduction has been rewritten, and the stated aims and associated relevance are no longer separated.

P2 L5 say "be used to measure" instead of "indicate"?

The entire section has been rewritten, and the word "indicate" is no longer used in the context it was in the original manuscript.

P2 L6. What about other sources of debris on glaciers, e.g. what about extra debris from mass movements from valley sides / lateral moraines? And possibility, at least, of subglacial debris brought to surface by thrusting?

We changed the language to acknowledge other potential sources, by stating that debris is "predominantly rockfall from valley walls" and by adding additional references.

P2 L 33. Say 200 and 600 to be consistent with above.

Fixed.

P2 L33. Say ". . .Glacier, Nepal. . ."

Fixed.

P2 L35 – P3 L 6. This reads like results / discussion. Suggest move from here to later.

Exception could be point about inability to drag antennas.

This is no longer in the Introduction.

Why is use of low frequencies "irrelevant"? Why is frequency relevant or not for areal coverage?

We removed this link between frequency and area to avoid confusion. The thought behind this statement was that a lower frequency GPR system could not, practically, be run from a drone (which would have vast areal coverage).

P3 L14-16. Something not right about this sentence. Also it presents results again. We need only a priori justifications for the methods. Or some reference to "preliminary data collection and analysis" or some such.

We added the following sentence to our Introduction: "With the initial observation of no distinct, detected debris-ice boundary, we sought a technique to measure debris thickness from GPR data lacking an interface delineation."

P3 L16. Should hypothesized be present tense?

We use the past tense when discussing past actions or decisions; we use present tense for discussion of the data and processing. We would be happy to change this at the discretion of the editor.

P3 L17. What do you mean by "favorable"? For what? Do you mean "high"?

What we meant was a medium likely to produce backscatter in penetrating electromagnetic waves; however, this language has been replaced.

P3 L18-19. This seems to be the key a priori reason for your approach. Is it the case that clasts are large at Changri Nup and dragging antenna is difficult, therefore you need a different approach involving use of backscatter information? I think you need a better articulation of the limitations of previous work, the differences between CN glacier cf. other glaciers studied, and therefore a better justification for your work

culminating in a clear set of aims.

In the Introduction, we summarize the reasoning for our approach: We explore the middle ground: GPR of a high frequency that could potentially be deployed on an airborne platform. We collected ground based radar measurements but did not detect a debris-ice interface. This differed from other studies, perhaps because of differences in debris lithology, porosity, or moisture content. Additionally, we clearly state our aim, which was to measure the thickness of Changri Nup's debris and, in doing so, explore how a 960 MHz system (i.e., a frequency relevant to remote radar systems, which operate near âĹij 1 GHz) performs in glacial debris. To this end, we compared the depth of volumetric backscatter from GPR data with manual ground truth measurements of debris thickness. We validate our indirect backscatter method with experimental studies and, thus, propose a methodology for determining debris thickness from GPR data that, for any number of reasons, do not show a clear debris bottom.

P4. Fig 1 etc. the term "cross" to refer to the transects running up and down glacier seems odd to me. I think of "cross glacier" and "cross transects" as going across a glacier and "along glacier" or "longitudinal transects" as going up and down glacier.

We have revised descriptions of the transects from "longitudinal" and "transverse" to "along-glacier" and "across-glacier."

P5. Last sentence of Fig Heading. Is what is stated really obvious from the photos? Doesn't appear so to me.

No, not obvious. So we have added descriptions of each of the panels of the figure to the main text.

P6. L5. Consider "Although, as shown later, this raising caused..."

This sentence was rewritten to read: "Although neither elevation was sufficient to separate the direct coupling (DC) and the return from the debris surface..."

P7. L10. Why is N  M 2017 referenced here?
The reference was for "sub-debris glacier surface topography is unknown (Nicholson and Mertes, 2017)." However, this has been removed from the rewrite.

P7. L14. "local" to what?

We now clarify the meaning as being local to the campus of Dartmouth College.

P7. L25. This sentence doesn't make sense to me.

This sentence has been removed.

P7. L28. Suggest mention things here in text that are currently in the Fig Heading.

Details from the caption have been moved to the text.

Also, important to explain the "surface reflection" is debris surface not ice surface.

We specified that the "surface reflection" is the "debris surface reflection."

P8 Fig 4 Caption. Change "Schematic" to "Annotated photograph"?

Fixed.

P9. Poor text quality in Fig 5. Y axis. In Fig caption suggest change sentence to something like: "In none of the stacked or uncompressed data profiles are the hyperbolic diffractions wide enough to be used to calculate the dielectric permittivity."

The figure has been remade and its caption shortened. This sentence is now in the main text and reads: "in neither the uncompressed (Figure 5) nor stacked (Figures 6 and 7) data for any profile are there hyperbolic diffractions sufficiently wide to be interpreted accurately for dielectric permittivity (Arcone, 1996; Yilmaz, 1987)."

P 9. L6. Could delete the glacier name here.

Fixed.

Also, is this sentence correct? The profiles show an absence of 3 things? Seems odd. Wouldn't it be best to state what they do show?

For clarity, we have now enumerated what the profiles show.

P9 L9. Suggest replace "dramatically" with "much".

We removed the word "dramatically."

P9 L16 "raises"

See above response on verb tenses.

P11. L9. Delete apostrophe after 'scatterers'.

This sentence was rewritten, and there is no longer the need for a possessive.

P12. Fig 8 Heading. Need consistency in referring to colours in key. Either refer to them all or none. Delete "climber's left to climber's right". Suggest move penultimate sentence to higher up to discuss all the lines in the Fig. first.

We think it necessary to include "climber's left to climber's right" in the description of the transects because the glacier does not flow in a straight line and describing the direction as "West to East" does not apply on all transects. We reordered the caption to describe the lines and symbols in the figure first.

P13 Fig 9. Explain the reason for the gaps in the traces. Same with Figure B2 in Appendix. P13. Fig 10 Heading and Fig 6 Heading earlier.

The gaps are explained in the captions for Figure 6 (formerly 8) and B2, the only times they appear. Former Figure 6 has been moved to the Supplementary Material, and details have been moved from the caption to the text. Information from the former Figure 10's caption has, similarly, been moved to the main text of the manuscript.

Need to better explain what "surface reflection" is. You have "surface reflection" but also "debris surface reflection". Surface reflection is abbreviated SR and S inconsistently. Why is bottom reflection not defined in Fig 6?

We have revised this figure in the new version of the manuscript to accommodate these

changes.

P14 L 1. "pine-board" ?

Here we are referring to both the pine boards and the pine shavings, so adding 'board' would alter that meaning.

P14. L2-4. How do these sentences follow on from the previous sentence? I don't understand what the evidence for this is.

These sentences build the argument that it is reasonable not to see a reflection at the debris-ice interface. We have added references for the permittivity of ice.

P14 L10-12. What is the evidence for this? Refer to a Figure and describe / label the Figure?

We have moved the description of the antenna height to a more appropriate location in the Methods section. Raising the antenna insufficiently (inadvertently) to separate out the DC from the returns did enhance the chances of detecting a signal from the debris-ice interface because the antenna height produced a far field spherical wave, the curvature of which approximated a plane when intersecting the surface. This is not an observation from our results, as a reader may have concluded from the original manuscript, but rather a property and result of the antenna geometry.

P14 L13. "To estimate. . ."

Changed to "For locating. . ."

P14 L14-15. What does this mean? Are you just using the measurements to calibrate an equation to determine depth from radar data?

Yes, this is a nutshell version of the method, which is explained further and clarified in the revised manuscript.

P14 L 15. The procedure needs to be explained wrt Fig 11. P14 L17. Threshold tao

Interactive
comment

needs explaining. You're referring to a threshold in the P14 L17-18. Confusion here. What is an iteration? If it's the number of times the "model" is run then it's run n-1 times right because you leave one out each time? P14 L19. ". . .assuring generation of quality statistics". Poor English.

All 4 of the above comments have been addressed by a careful rewrite of this section.

P14 L 22. Can't Figs 11 and 12 be combined? Could use colors to show 20, 30, 40, etc

We have eliminated previous Figure 11 from the revised manuscript. We could indeed add colors on the curve to show percentages; though we think this might be more complicated for many readers, so we have kept the simplicity of the original figure 12.

P15 L5-6. Suggest "There is broad-scale agreement between the calculated and measured average depths (Table 3). An exact match is not expected because:. . ."

This is addressed by the language in the revised manuscript.

P16 L3. Delete "field measurements of solid block)" I assume?

We cite both Hubbard et al (1997) and our own field measurements for the claim that granite has a dielectric between 5 and 7. We switched the order of these citations to avoid confusion.

P16 L 5. Suggest ". . .shallow depths. This, together with point (a) above, emphasize. . ." P16 L11. Are you referring to debris grain size here? Small and uniform?

The above 2 comments are addressed by the rewritten language in the revised manuscript.

P17. L3. "would" not "should" to be consistent with L2.

Fixed.

P17 L20. ". . .lack to debris too thin, debris too thick, and high scatter" Poor English.

Rewrite.

Rewritten.

P17 L21. ". . .but fail for thin layers". What does this mean?

Eliminated from the revised manuscript.

P18 L56. No need for new paragraph here as you're discussing the same point.

P18 L30. These numbers here are different to those in Table 3.

Fixed.

Why only refer to the 3 cross glacier transects rather than the other 2 here?

We explained that lack of ground-truth points and the short length of these transects makes use of them more difficult.

P18 L31. It is really the case that these thicknesses "do not vary significantly"?

We have now put the variability in thickness with thresholds calculated locally into Table 3, showing how the change in threshold percent gives a change in depth. The changes in depth remain smaller than the uncertainty.

P20 L7. ..."dielectric permittivity across..."

This has been eliminated in the rewrite.

P21-32. Appendixes have got mixed up with the references. Might be useful to show location and look direction of all the photos on a Map (e.g. Fig 1).

The formatting (references mixed in with appendix figures) has been fixed. Since the photos are mainly intended to show the terrain (and debris-surface variability), we think a map is not necessary.

---

## Author Comment (AC3) · 21 Oct 2019

Reviewer 3 (Anonymous)'s REVIEWS ON TCD VERSION

Page 3, Line 10: "...low frequencies irrelevant for efficient areal coverage..." There is no reason why the systems used in the cited studies (and much larger systems) could not be deployed by helicopter (I have a helicopter-slung system more than 20 m long for example).

Fair point, we have stricken this language from the revised text.

[Figure]

"...dragged antennas, an approach that was impossible at our field site..." The surface of Ngozumpa, Lirung and Langtang glaciers are very similar to the pictures of Changri Nup shown here.

Fair enough. We would have gladly used an antenna-dragging approach at Changri Nup if it seemed possible, but it did not.

Page 3, Line 26: "frequency relevant to remote systems..." by which you mean drones. See above comment regarding helicopters, which are readily available for charter in the Khumbu.

We have stricken the relevancy language.

Page 4, Line 11: The frozen ice surface is critical here. By far the largest control on the dielectric contrast in such a setting is the presence or absence of water. It would be much easier to detect the ice-debris interface if you surveyed in dry weather by in thawing conditions (so debris is largely dry but ice surface is wet). This is what McCarthy et al. and Nicholson et al. did.

This is true, and in our Discussion we added that "in the ablation season, [other authors] likely encountered a wet interface, a 'saturated layer' of water on ice, below mostly dry debris. A wet interface gives a much stronger reflection than a dry one." However, it is not possible to repeat our field season in late Spring or Summer; were it possible, we would then have to consider the logistics of a potential snow cover on top of the debris and potential saturated debris (water saturated debris would attenuate the radar signal and, thus, introduce a further complication).

Page 9, Line 3: "Profiles at 50 ns and 100 ns..." Not sure what this means, please clarify.

By this, we meant that the profiles taken over the two along-glacier transects were collected with one of two different time range settings: 50 ns and 100 ns. We have updated the language to "profiles collected at time ranges of 50 ns and 100 ns..." to

clarify.

Page 9, Line 8 and in general: I don't think the term 'volume scatter' is used correctly. It is normally used to describe the net result of multiple bounces from many discrete, closely-spaced reflectors of similar size to the wavelength, within some medium, which I think is what you have here. In such a case, the interfaces are likely to be air-rock, rock-rock and ice-rock interfaces. I think you're describing it as single-bounce reflections back from individual point scatterers within the debris layer, which is analogous to detecting an aircraft in flight (which clearly is not volume scatter). If it was dominated by single-point scatterers then you'd expect to see hyperbolae in the unstacked radargram (e.g., Figure 5) resulting from the radar moving closer to then further away from the point scatterer, but I don't see these. It's also not clear why the signal would penetrate through the debris above but reflect from these particular buried scatterers.

We agree on the definition of "volume scatter." The language in question that caused confusion over our use of the term has been removed in the rewrite.

Page 9, Line 9: "The volumetric backscatter is dramatically stronger than the surface return..." But the surface return is obscured/interfered with by the direct wave, so it is not clear what the surface return strength is from the field data. Indeed, the lab experiments show the surface return at least as strong as the volume scatter (Figure 10).

That's entirely correct, we can't say this for sure. We have deleted this statement from the revised version.

Page 11, Rock box experiments: these are really nice and I'd like to see more, with the aim of characterising the attenuation and ice-surface detectability. I suggest: Increasing the debris thickness (round trip of 57 cm is thickness of only 28.5 cm, which doesn't capture the potential range of thicknesses in the field). Vary the thickness and measure how the aluminium-base signal strength varies, to characterise attenuation with depth. With the pine base, try wetting the pine (by pouring water in at the base,

while keeping the debris largely dry) to simulate a thawing ice surface. Try the above but wetting the debris from above, to simulate rain or snowmelt. Repeat the above with a range of GPR frequencies.

These are great ideas and could form the basis of a whole paper in itself, but we feel they are well beyond the scope of this manuscript. The point of the rock box experiments was mainly to see (1) whether signals did penetrate at least to the average depth and the dryness found in the field and (2) whether that produced a bottom reflection. We clarified the aims of our study, including the purpose of the rock box experiments, in our rewritten Introduction. We also state explicitly in the Conclusions that future experiments should approach the depth problem.

Page 14, Line 14: "...area under the Hilbert transformed curve..." – meaning that the value calculated is the sum of the gain-corrected backscatter power over the chosen time window?

For estimating the depth of the debris-ice interface, we used the progressive, integrated area as a function of time under the gain-corrected, Hilbert transformed trace. This progressive area was taken as a measure of the total backscattered power at that antenna location, from the depth equivalent of the time range, using a dielectric of 3 for debris. We realize that the inherent beamwidth of the antenna allowed many backscattered events, including that of the surface reflection and the DC, to interfere, but, on average, the progression of this integrated area represented a measure of volumetric backscatter. This has been clarified in the manuscript.

This would seem to depend on both the gain function used and on the time-window selected.

That is correct. We used a spherical correction to approximate the actual loss caused by beam spreading. And, as the window lengthens, the area of integration under the trace increases for each unit of time.

[Figure]

As the signal attenuates with depth, eventually it reaches the noise floor (i.e., system noise), so if a long time window was chosen for exactly the same profiles, this would add 'power' to the summed magnitude because the this approach would sum up this noise. This would increase the total summed power and so the 38

Also, I think this approach assumes a linear relationship between summed backscatter power and debris thickness, but the effect of attenuation would mean that this is incorrect.

We used a geometric spherical beam spreading gain correction. Consequently, the deeper the returns, the greater the integrated area. Thus, we might expect that energy of return increase with depth. But it does not because the returns from deeper in the debris, for the most part, have been scattered away from the antenna direction. A linear relationship is what one might get if we had a pencil beam, like an X-ray in a CT scan of uniform material. Given that attenuation caused by scattering is complex, the LOOCV method is reasonable in that it makes no assumption of any such relationship.

This is because the near-surface returns contribute much more to the sum than the deeper ones, so for example, doubling the debris thickness would not double the summed backscatter power, the increase would be much less than double. I think the sensitivity of the summed power to increasing thickness would follow a decay curve.

Yes, the near surface returns do make a stronger contribution, and, yes, doubling the thickness would not double the summed backscatter power. However, our statistical approach is relatively insensitive to this nonlinearity. We have clarified this idea in the text.

Clearly the deeper debris should contribute less backscatter because much of the transmitted signal has already been scattered back by the debris above, and so is no longer available.
In most cases, it is only a small part of the signal that is backscattered because the presence of coherent pulses implies that single scattering is present. Thus, most of the energy that encountered a single rock went around the rock and just a small fraction was backscattered. The in situ wavelength for a 960 MHz signal is 31 cm/1.73 = 18 cm, which is much bigger than most of the clasts in the rock box and, as explained in the text, probably most of the clasts we encountered in our profiles on Changri Nup glacier. In summary, the nature of backscatter from rough debris combined with the wavelength we used presented a complicated situation in which the threshold method was a valid statistical approach. All of the profiles show single scattering and suggest that the loss of energy with depth was likely caused by deeper and deeper backscatter that did not reach the antenna (not by backscatter that was attenuated because of losses higher up in the debris layer).

This also means that the variation in the threshold calculated locally (e.g., the 35

Admittedly, there is some nonlinearity. We have now put the variability in thickness with thresholds calculated locally into Table 3, showing how the change in threshold percent gives a change in depth. The changes in depth remain smaller than the uncertainty.

A further implication of attenuation is that there is certainly a limit to the debris thickness that can be quantified by GPR (page 18, line 1, "...we do not have a thickness limitation.") – above a certain thickness, there is no longer enough signal for backscatter to be detected, and therefore no sensitivity to debris. It is not obvious a priori what this limit is because it depends on the debris properties, but once this is exceeded than all that can be said is that the debris is thicker than this detectability limit.

We have changed the language about our thickness limitation to show that our approach does not have such a thickness limitation within the range of debris thicknesses likely to be encountered in a supraglacial setting.

Finally, the 38% threshold is empirical and so does depend on the local properties (of porosity, lithology, grain size distribution, wetness). This means that it may be useful

for interpolating thicknesses within a single survey, but is unlikely to be universally applicable (as suggested later).

We remove the assertion that 38% can be universally applied in our updated manuscript. Rather, we state in the Conclusions that the 38% threshold that matches debris thicknesses on Changri Nup Glacier is not completely transferable to other glaciers but, nevertheless, may indicate debris thickness on layers with mineralogy and porosity similar to Changri Nup's. Future work could assess the transferability of the specific threshold on other glaciers.

Page 14, Line 24: If the debris layer produces 38% of the backscatter, where does the other 62% of backscatter come from? Seems that you've already ruled out significant backscatter from the surface and sub-debris ice.

Page 16, Line 13 and elsewhere: be careful with the use of the word 'coherence' – it has a particular meaning in radar processing, whereas I think you're using it to mean 'detectable and continuous' or similar.

We have been more careful use of this term in the revision.

Page 16, Line 29: need to add these porosity uncertainties into the debris-thickness error budget (yellow lines in Figure 8 etc).

We believe it most appropriate to represent uncertainty in Figures 6, 7, A1, A2 and in Table 3 consistently and with the single metric of average RMSE from the LOOCV. However, it was a good point that this reviewer made, and we enhanced the Uncertainty section to include a discussion of adjustments to the depth scale caused by porosity differences/assumptions. We recalculated the threshold with a porosity of 30% and showed that it did not change much. We added to our Conclusions a clarification that our presented threshold percentage is specific to the porosity and mineralogy of the debris we measured. The 38% threshold that matches debris thicknesses on Changri Nup Glacier is not totally transferable to other glaciers but, nevertheless, may indicate

debris thickness on layers with mineralogy and porosity similar to Changri Nup's. Future work could assess the transferability of the specific threshold on other glaciers.

Page 17, Line 11: by far the greatest influence on ice detectability in at least the McCarthy and Nicholson studies is the wet ice surface.

We acknowledge this in the revised manuscript. In our Discussion, we added that in the ablation season, [other authors] likely encountered a wet interface, a 'saturated layer' of water on ice, below mostly dry debris. A wet interface gives a much stronger reflection than a dry one.

Section 4.3: see above regarding volume scatter versus point scatter.

We agree on the definition of "volume scatter." The language in question that caused confusion over our use of the term has been rewritten or removed in the rewrite.

---

## Author Comment (AC4) · 6 Mar 2020

We want to thank this reviewer for his/her time, as the feedback helped improve the manuscript considerably. We realized in preparing the next upload that two responses were inadvertently left out of the letter on October 21, 2019 due to copy/paste mistakes. Below is what we intended for the letter, with inadvertently omitted passages in bold. The remainder is identical.

Also of note is that since October, we, as stated in our March 2020 Response Letter, no longer use the threshold corresponding to the single lowest RMSE from the LOOCV but rather the mean threshold from the lowest n RMSEs, where n is the number of ground-truth measurements along that transect. We subtract the antenna elevation during the thickness retrieval calculation. This occasionally yields negative, non-physical thicknesses; we discard these negative values but report their occurrence in Table 3. Using this more robust method, we find a threshold of 42% for a 19-cm elevated antenna and 53% for a 27-cm one. Because the applied gain function places the deeper returns at higher gain in the data collected with a 27-cm elevated antenna, we do not combine these numbers and, instead, assert that 42% is our threshold finding since 4 of 5 transects were measured with an antenna elevated by 19 cm.

Page 3, Line 10: "...low frequencies irrelevant for efficient areal coverage..."
There is no reason why the systems used in the cited studies (and much larger systems) could not be deployed by helicopter (I have a helicopter-slung system more than 20 m long for example).
Fair point, we have stricken this language from the revised text.
"...dragged antennas, an approach that was impossible at our field site..."
The surface of Ngozumpa, Lirung and Langtang glaciers are very similar to the pictures of Changri Nup shown here.
Fair enough. We would have gladly used an antenna-dragging approach at Changri Nup if it seemed possible, but it did not.
Page 3, Line 26: "frequency relevant to remote systems..." by which you mean drones. See above comment regarding helicopters, which are readily available for charter in the Khumbu.
We have stricken the relevancy language.
Page 4, Line 11: The frozen ice surface is critical here. By far the largest control on the dielectric contrast in such a setting is the presence or absence of water. It would be much easier to detect the ice-debris interface if you surveyed in dry weather by

in thawing conditions (so debris is largely dry but ice surface is wet). This is what McCarthy et al. and Nicholson et al. did.

This is true, and in our Discussion we added that "in the ablation season, [other authors] likely encountered a wet interface, a 'saturated layer' of water on ice, below mostly dry debris. A wet interface gives a much stronger reflection than a dry one." However, it is not possible to repeat our field season in late Spring or Summer; were it possible, we would then have to consider the logistics of a potential snow cover on top of the debris and potential saturated debris (water saturated debris would attenuate the radar signal and, thus, introduce a further complication).

Page 9, Line 3: "Profiles at 50 ns and 100 ns..." Not sure what this means, please clarify.

By this, we meant that the profiles taken over the two along-glacier transects were collected with one of two different time range settings: 50 ns and 100 ns. We have updated the language to "profiles collected at time ranges of 50 ns and 100 ns..." to clarify.

Page 9, Line 8 and in general: I don't think the term 'volume scatter' is used correctly. It is normally used to describe the net result of multiple bounces from many discrete, closely-spaced reflectors of similar size to the wavelength, within some medium, which I think is what you have here. In such a case, the interfaces are likely to be air-rock, rock-rock and ice-rock interfaces. I think you're describing it as single-bounce reflections back from individual point scatterers within the debris layer, which is analogous to detecting an aircraft in flight (which clearly is not volume scatter). If it was dominated by single-point scatterers then you'd expect to see hyperbolae in the unstacked radargram (e.g., Figure 5) resulting from the radar moving closer to then further away from the point scatterer, but I don't see these. It's also not clear why the signal would penetrate through the debris above but reflect from these particular buried scatterers.

We agree on the definition of "volume scatter." The language in question that caused confusion over our use of the term has been removed in the rewrite.

Page 9, Line 9: "The volumetric backscatter is dramatically stronger than the surface return..." But the surface return is obscured/interfered with by the direct wave, so it is not clear what the surface return strength is from the field data. Indeed, the lab experiments show the surface return at least as strong as the volume scatter (Figure 10).

That's entirely correct, we can't say this for sure. We have deleted this statement from the revised version.

Page 11, Rock box experiments: these are really nice and I'd like to see more, with the aim of characterising the attenuation and ice-surface detectability.

I suggest: Increasing the debris thickness (round trip of 57 cm is thickness of only 28.5 cm, which doesn't capture the potential range of thicknesses in the field).

Vary the thickness and measure how the aluminium-base signal strength varies, to characterise attenuation with depth.

With the pine base, try wetting the pine (by pouring water in at the base, while keeping the debris largely dry) to simulate a thawing ice surface.

Try the above but wetting the debris from above, to simulate rain or snowmelt. Repeat the above with a range of GPR frequencies.

These are great ideas and could form the basis of a whole paper in itself, but we feel they are well beyond the scope of this manuscript. The point of the rock box experiments was mainly to see (1) whether signals did penetrate at least to the average depth and the dryness found in the field and (2) whether that produced a bottom reflection.

We clarified the aims of our study, including the purpose of the rock box experiments, in our rewritten Introduction. We also state explicitly in the Conclusions that future experiments should approach the depth problem.

Page 14, Line 14: "...area under the Hilbert transformed curve..." – meaning that the value calculated is the sum of the gain-corrected backscatter power over the chosen time window?

For estimating the depth of the debris-ice interface, we used the progressive, integrated area as a function of time under the gain-corrected, Hilbert transformed trace. This progressive area was taken as a measure of the total backscattered power at that antenna location, from the depth equivalent of the time range, using a dielectric of 3 for debris. We realize that the inherent beamwidth of the antenna allowed many backscattered events, including that of the surface reflection and the DC, to interfere, but, on average, the progression of this integrated area represented a measure of volumetric backscatter. This has been clarified in the manuscript.

This would seem to depend on both the gain function used and on the time-window selected.

That is correct. We used a spherical correction to approximate the actual loss caused by beam spreading. And, as the window lengthens, the area of integration under the trace increases for each unit of time.

As the signal attenuates with depth, eventually it reaches the noise floor (i.e., system noise), so if a long time window was chosen for exactly the same profiles, this would add 'power' to the summed magnitude because the this approach would sum up this noise. This would increase the total summed power and so the 38% level would be deeper. This would be a spurious increase, unrelated to the actual debris thickness.

**Good point. For this reason, we do not assume that this threshold value (38% in our study) is valid on all glaciers or even on Changri Nup with a different GPR system, and we have removed any implication along these lines from the text. Fundamentally, this paper develops a method and presents a guide for how to conduct a study to discern debris thickness from GPR data when and where an interface is not detected. In the present case, because our data appeared to show a reduction in backscatter at a depth consistent with the expected debris-ice interface, we sought to find the interface algorithmically, by the reduction of backscatter.**

Also, I think this approach assumes a linear relationship between summed backscatter power and debris thickness, but the effect of attenuation would mean that this is incorrect.

We used a geometric spherical beam spreading gain correction. Consequently, the deeper the returns, the greater the integrated area. Thus, we might expect that energy of return increase with depth. But it does not because the returns from deeper in the debris, for the most part, have been scattered away from the antenna direction. A linear relationship is what one might get if we had a pencil beam, like an X-ray in a CT scan of uniform material. Given that attenuation caused by scattering is complex, the LOOCV method is reasonable in that it makes no assumption of any such relationship.

This is because the near-surface returns contribute much more to the sum than the deeper ones, so for example, doubling the debris thickness would not double the summed backscatter power, the increase would be much less than double. I think the sensitivity of the summed power to increasing thickness would follow a decay curve.

Yes, the near surface returns do make a stronger contribution, and, yes, doubling the thickness would not double the summed backscatter power. However, our statistical approach is relatively insensitive to this nonlinearity. We have clarified this idea in the text.

Clearly the deeper debris should contribute less backscatter because much of the transmitted signal has already been scattered back by the debris above, and so is no longer available.

In most cases, it is only a small part of the signal that is backscattered because the presence of coherent pulses implies that single scattering is present. Thus, most of the energy that encountered a single rock went around the rock and just a small fraction was backscattered. The in situ wavelength for a 960 MHz signal is 31 cm/1.73 = 18 cm, which is much bigger than most of the clasts in the rock box and, as explained in the text, probably most of the clasts we encountered in our profiles on Changri Nup glacier. In summary, the nature of backscatter from rough debris combined with the wavelength we used presented a complicated situation in which the threshold method was a valid statistical approach. All of the profiles show single scattering and suggest that the loss of energy with depth was likely caused by deeper and deeper backscatter

that did not reach the antenna (not by backscatter that was attenuated because of losses higher up in the debris layer).

This also means that the variation in the threshold calculated locally (e.g., the 35%, 42%, 43% in Table 3) could mean large variability in thickness, i.e. including another 8% of the summed power could mean including a lot more debris, because the debris at depth contributes little to the total.

Admittedly, there is some nonlinearity. We have now put the variability in thickness with thresholds calculated locally into Table 3, showing how the change in threshold percent gives a change in depth. The changes in depth remain smaller than the uncertainty.

A further implication of attenuation is that there is certainly a limit to the debris thickness that can be quantified by GPR (page 18, line 1, "...we do not have a thickness limitation.") – above a certain thickness, there is no longer enough signal for backscatter to be detected, and therefore no sensitivity to debris. It is not obvious a priori what this limit is because it depends on the debris properties, but once this is exceeded than all that can be said is that the debris is thicker than this detectability limit.

We have changed the language about our thickness limitation to show that our approach does not have such a thickness limitation within the range of debris thicknesses likely to be encountered in a supraglacial setting.

Finally, the 38% threshold is empirical and so does depend on the local properties (of porosity, lithology, grain size distribution, wetness). This means that it may be useful for interpolating thicknesses within a single survey, but is unlikely to be universally applicable (as suggested later).

We remove the assertion that 38% can be universally applied in our updated manuscript. Rather, we state in the Conclusion that the 38% threshold that matches debris thicknesses on Changri Nup Glacier is not totally transferable to other glaciers but, nevertheless, may indicate debris thickness on layers with mineralogy and porosity similar to Changri Nup's. Future work could assess the transferability of the specific

threshold on other glaciers.

Page 14, Line 24: If the debris layer produces 38% of the backscatter, where does the other 62% of backscatter come from? Seems that you've already ruled out significant backscatter from the surface and sub-debris ice.

**The LOOCV suggested that the time by which 38% of the integrated energy has been scattered corresponds to the depth of the debris layer. While this does mean 62% of the integrated energy is scattered elsewhere, the threshold is highly sensitive to the number of samples we used for profiling: 1024 (corresponding to ∼4 m depth). As is clear in the appendix Figures A1 and A2 which show all recorded data, most energy is scattered in the near-surface. The subset of Figure 9 shows the same trace to its full recorded depth; each scan, with 1024 samples, has a long, lumpy tail that integrates to a majority of the energy because of its length. Had we used 256 samples, which would have still penetrated the interface in all cases, the threshold instead would have been 65%. While we cannot rule out the possibility of significant scattering elsewhere, the depth integrated scatter is high in the debris region, and 38% over 1024 samples is where the integrated energy corresponds to the debris thickness. We added a paragraph to the "Leave-one-out cross validation" section to clarify this.**

Page 16, Line 13 and elsewhere: be careful with the use of the word 'coherence' – it has a particular meaning in radar processing, whereas I think you're using it to mean 'detectable and continuous' or similar.

We have been more careful use of this term in the revision.

Page 16, Line 29: need to add these porosity uncertainties into the debris-thickness error budget (yellow lines in Figure 8 etc).

We believe it most appropriate to represent uncertainty in Figures 6, 7, A1, & A2 and in Table 3 consistently and with the single metric of average RMSE from the LOOCV. However, it was a good point that this reviewer made, and we enhanced the Uncertainty section to include a discussion of adjustments to the depth scale caused

by porosity differences/assumptions. We recalculated the threshold with a porosity of 30% and showed that it did not change much. We added to our Conclusions a clarification that our presented threshold percentage is specific to the porosity and mineralogy of the debris we measured: "The 38% threshold that matches debris thicknesses on Changri Nup Glacier is not totally transferable to other glaciers but, nevertheless, may indicate debris thickness on layers with mineralogy and porosity similar to Changri Nup's. Future work could assess the transferability of the specific threshold on other glaciers."

Page 17, Line 11: by far the greatest influence on ice detectability in at least the McCarthy and Nicholson studies is the wet ice surface.

We acknowledge this in the revised manuscript. In our Discussion, we added that "in the ablation season, [other authors] likely encountered a wet interface, a 'saturated layer' of water on ice, below mostly dry debris. A wet interface gives a much stronger reflection than a dry one."

Section 4.3: see above regarding volume scatter versus point scatter.

We agree on the definition of "volume scatter." The language in question that caused confusion over our use of the term has been rewritten or removed in the rewrite.